# Locret: Enhancing Eviction in Long-Context LLM Inference with Trained Retaining Heads on Consumer-Grade Devices

**Yuxiang Huang**[*]                                                              *huang-yx21@mails.tsinghua.edu.cn*
*Dept. of Computer Science and Technology, BNRIST, IAI, Tsinghua University, Beijing, China.*

**Binhang Yuan**[†]                                                                      *biyuan@ust.hk*
*Department of Computer Science and Engineering, HKUST, Hong Kong, China.*

**Xu Han**[†]**, Chaojun Xiao, Zhiyuan Liu**                         *{han-xu, xcj, liuzy}@tsinghua.edu.cn*
*Dept. of Computer Science and Technology, BNRIST, IAI, Tsinghua University, Beijing, China.*

**Reviewed on OpenReview:** *https://openreview.net/forum?id=YPVBCTBqHE*

## Abstract

Scaling the input context length of a large language model (LLM) incurs a significant increase in computation cost and memory footprint to maintain the attention key-value (KV) cache. Existing KV cache compression methods suffer from inefficient compression strategies and limited memory reduction effects, making it difficult for LLMs to conduct long-context inference on consumer-grade devices, especially when inferring long-context stream input. Such obstacles prevent consumer-grade devices from supporting more complex applications, creating challenges for the democratization of LLMs. To overcome this, we propose LOCRET, a framework to create an eviction policy compatible with chunked prefill. By evaluating the causal importance of KV cache units using *retaining heads*, LOCRET enables precise eviction of cache units, facilitating efficient long-context inference. In our empirical studies, LOCRET outperforms the recent popular and competitive approaches in terms of memory efficiency and generation quality — LOCRET achieves up to $20\times$ of KV cache compression ratio within less than 10% performance loss. Furthermore, LOCRET achieves 128K+ long-context inference on a single NVIDIA 4090 GPU without compromising generation quality and only costs $< 1$ GPU hour of additional training.

## 1 Introduction

In recent years, large language models (LLMs) have revolutionized generative AI (Zhao et al., 2023; Minaee et al., 2024), and the advancements of LLMs in handling long-context tasks have further unlocked the potential of generative AI. As a result, the context lengths supported by state-of-the-art LLMs have been significantly extended, such as GPT-4o (OpenAI, 2024) handling 128K tokens, Claude-3 (Anthropic, 2024) supporting 200K tokens, and Gemini-1.5 (Reid et al., 2024) even reaching 10M tokens. These improvements enable LLMs to tackle complex applications with extremely long or streaming inputs, such as multi-hop reasoning (Li et al., 2024a; Schnitzler et al., 2024), LLM-driven agents (Qin et al., 2024b; Wang et al., 2024), and AI-powered operating systems (Mei et al., 2024). Some recent efforts (Hu et al., 2024b; Abdin et al., 2024) have successfully deployed LLMs on consumer-grade end-side devices instead of cloud servers and conducted inference with limited context. We envision that unleashing the potential of long-context inference on consumer-grade devices will revolutionize the development of personalized AI applications and the democratization of LLMs. However, *conducting long-context LLM inference on consumer-grade devices remains a challenging problem that requires algorithmic innovations and systematic optimizations.*

---

[*]Work done during internship at HKUST.
[†]indicates corresponding authors.

As context length scales, the challenge of long-context LLM inference arises from two major aspects: the increased computational cost of the attention mechanism and the higher memory footprint due to the key-value (KV) cache. This leads to the failure of traditional optimizations targeting model backbones to provide a sufficient solution. Specifically, backbone-targeted optimizations, such as compacting model architectures (Hu et al., 2024b; Abdin et al., 2024) and quantizing model weights (Frantar et al., 2023; Dettmers et al., 2022; Xiao et al., 2023; Lin et al., 2024), fail to improve the efficiency of attention patterns or the KV cache, as attention's quadratic complexity concerning sequence length remains unaddressed. To this end, recent efforts target optimizing attention patterns and the KV cache to achieve efficient long-context inference.

Recent attention-targeting optimizations, including sparse attention (Jiang et al., 2024a; Ge et al., 2024; Lou et al., 2024) and KV cache quantization (Liu et al., 2024b; Hooper et al., 2024; Zandieh et al., 2024), show promising results to accelerate attention computation and reduce memory footprint. However, they fail to fundamentally address the core challenge: *the KV cache grows linearly with context length*. Layer-wise chunked prefill combined with attention sparsity (e.g., SNAPKV (Li et al., 2024b)) can alleviate this problem to a certain extent. This technique typically performs cache eviction after the precise attention computation for each layer and requires access to the entire sequence. It can theoretically support longer sequences by limiting the maximum memory usage to a single layer's KV cache, but *it cannot handle streaming input whose length grows continually*. The combination of cache eviction methods (Xiao et al., 2024c; Yang et al., 2024) and chunked prefill offers a more promising approach by maintaining a static cache size and supporting streaming input. Yet, existing eviction techniques like $H_2O$ (Zhang et al., 2024d) and SNAPKV show significant discrepancies between local importance estimation and global importance estimation, i.e., it is hard to estimate the importance of each token only based on its previous tokens. Instead, these methods require a large number of subsequent tokens to make an accurate estimation. Other methods like SIRLLM (Yao et al., 2024) show great local-global estimation consistency but suffer from performance degradation. We present a detailed analysis of the background in Section 3.2.

Here, we list our contributions below:

**Contribution 1:** We propose LOCRET, a lightweight training-based paradigm for selective KV cache eviction in long-context LLM inference. It introduces learnable *retaining heads* to estimate the *causal importance score* (CIS) for token selection, with an offline training cost of <1 GPU hour. Additionally, we present LOCRET-Q, a query-aware variant of LOCRET, slightly modified to handle query-driven tasks (e.g., retrieving a value of a given key in a long JSON string).

**Contribution 2:** We provide an efficient inference system implementation for LOCRET, integrating retaining heads into a chunked prefill framework. This integration limits GPU memory usage by evicting low-CIS KV cache units during the prefill process, thereby accelerating the prefill time. LOCRET is compatible with all decoder-only LLMs and imposes minimal additional hardware requirements.

**Contribution 3:** We extensively evaluate LOCRET, demonstrating its ability to achieve comparable performance with full KV cache while maintaining inference efficiency. LOCRET achieves over $20\times$ and $8\times$ KV cache compression ratios for `Phi-3-mini-128K` and `Llama-3.1-8B-instruct`, respectively. Additionally, LOCRET-Q accelerates prefill by over $2\times$ on query-driven tasks without significant performance degradation. This framework enables full comprehension of long contexts on consumer-grade devices without compromising the generation quality, and introduces minimal additional system optimizations.

## 2  Related Work

This paper focuses on optimizing long-context LLM inference. Existing efforts can be categorized into algorithm and system optimization. For more details about LLMs, please refer to the surveys (Zhao et al., 2023; Lu et al., 2024).

**Algorithm Optimizations** aim to reduce the size of the KV cache and can generally be classified into three categories: quantization-based methods, sparsity-based methods, and token dropping methods. Quantization-based methods (Liu et al., 2024b; Hooper et al., 2024; Zandieh et al., 2024; Zhang et al., 2024a) use low-bit values to represent the KV cache, reducing cache memory overhead and improving cache computing efficiency. These quantization-based methods suffer from hardware-oriented operator customization and

additional inverse quantization overhead. Sparsity-based methods (Ge et al., 2024; Jiang et al., 2024a; Yang et al., 2024; Lou et al., 2024; Lv et al., 2024; Xu et al., 2025) leverage the sparsity patterns of attention heads to reduce both computational and I/O costs. Combining different patterns can yield further optimization by identifying specific patterns for each head (Ge et al., 2024; Jiang et al., 2024a; Xiao et al., 2024b). Trainable sparse attention methods, such as NSA (Yuan et al., 2025) and MoBA (Lu et al., 2025), adapt sparsity patterns during training to improve task performance. For more details on sparsity-based methods, please refer to the surveys (Yuan et al., 2024; Kang et al., 2024; Shi et al., 2024). Although quantization-based methods and sparsity-based methods have achieved promising results, they cannot address the issue that the KV cache memory overhead increases linearly with the context length. Eviction-based methods, such as $H_2O$ (Zhang et al., 2024d), ScissorHands (Liu et al., 2024a), and SirLLM (Yao et al., 2024), rank KV cache units by certain statistical metrics to identify the most influential units, discarding others to reduce memory usage and speed up attention computation. InfiniPot (Kim et al., 2024) introduces context distillation combined with chunked prefill, enabling effective long-context processing even under memory-constrained conditions. Pooling-based methods (Nawrot et al., 2024; Rajput et al., 2024), especially StreamingLLM (Xiao et al., 2024c) and LoCoCo (Cai et al., 2024a), compress multiple adjacent KV cache units into a single unit using pre-designed transformations. More important units will merge into compressed units with higher weights. Eviction-based and pooling-based methods drop or merge tokens to maintain a static cache size, but struggle with accurate victim selection and optimal pooling function design.

**System Optimizations** alleviate the challenge of long-context inference from a system-level perspective, by fully considering hardware features. Offloading-based methods (Sheng et al., 2023; Xiao et al., 2024a; Wu et al., 2024; Sun et al., 2024) use CPU memory to store the KV cache and retrieve only the most relevant chunks to GPU memory before inferring a new chunk. These methods reduce maximum GPU memory usage at the cost of introducing CPU-GPU communication overhead. Hardware-aware methods, such as Flash-Atttention (Dao et al., 2022; Dao, 2024; Shah et al., 2024) and Page-Attention (Kwon et al., 2023), enable more efficient runtime memory management by considering GPU architectures (Ghorpade et al., 2012). In addition, building inference infrastructures with a more efficient programming language (llama.cpp; llama2.c; rustformers), or adopting disaggregated inference (Jiang et al., 2024b; Zhong et al., 2024; Qin et al., 2024a; Hu et al., 2024a), can also greatly improve long-context inference efficiency. Since system optimizations primarily enhance efficiency by leveraging hardware resources rather than directly optimizing attention patterns or the KV cache, relying solely on them cannot adequately address the challenges of long-context LLM inference. Several efforts have integrated algorithm optimizations into system optimizations (Agrawal et al., 2023; Lee et al., 2024), such as KTransformers (KVCache.AI, 2024) leveraging offloading based on InfLLM (Xiao et al., 2024a), and StarAttn (Acharya et al., 2025) and APB (Huang et al., 2025) integrate approximate attention with sequence parallelism to achieving faster inference.

## 3 Background

### 3.1 Preliminaries

**Transformer Architecture.** Given a token sequence $\{t_1, \cdots, t_n\}$ as the input prompt of the transformer-based LLM, we denote the output hidden states of the $i$-th layer as $\mathbf{H}^{(i)}$, and denote the input embeddings of the first layer as $\mathbf{H}^{(0)}$. For each transformer layer, it consists of an attention block and a feedforward neural network (FFN) block. Attention blocks often follow the grouped-query attention (GQA) architecture (Ainslie et al., 2023), with $h$ query heads and $h/g$ KV heads, where $g$ is the group size, i.e., $g$ heads share the same KV heads. The multi-head attention (MHA) architecture adopted in the original transformer can be regarded as a special GQA ($g = 1$). In the $i$-th layer, the attention score of the $j$-th query head is formalized as

$$
\begin{aligned}
\left[\mathbf{Q}_j^{(i)}, \mathbf{K}_{\lceil j/g \rceil}^{(i)}, \mathbf{V}_{\lceil j/g \rceil}^{(i)}\right] &= \mathbf{H}^{(i-1)} \cdot \left[\mathbf{W}_j^{(i)\mathbf{Q}}, \mathbf{W}_{\lceil j/g \rceil}^{(i)\mathbf{K}}, \mathbf{W}_{\lceil j/g \rceil}^{(i)\mathbf{V}}\right], \\
\mathbf{A}_j^{(i)} &= \mathtt{softmax}\left(\mathbf{M} \odot \mathbf{Q}_j^{(i)} \mathbf{K}_{\lceil j/g \rceil}^{(i)\top} / \sqrt{d_m}\right) \cdot \mathbf{V}_{\lceil j/g \rceil}^{(i)}
\end{aligned}
\tag{1}
$$

where $d_m$ represents the hidden size for each head. After obtaining the attention score, the output of the $i$-th attention block is $\mathbf{A}^{(i)} = \left[\mathbf{A}_1^{(i)}, \cdots, \mathbf{A}_h^{(i)}\right] \cdot \mathbf{W}^{(i)\mathbf{Q}}$, and the output hidden states is $\mathbf{H}^{(i)} = \mathtt{FFN}(\mathbf{A}^{(i)})$.

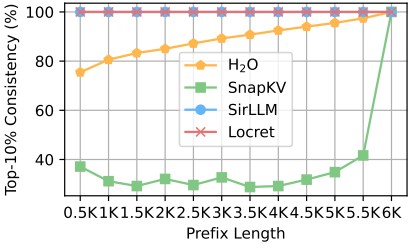

Figure 1: The $p_{0.1n}$ consistency.

Table 1: $\infty$Bench scores of $H_2O$, SNAPKV and LOCRET. $H_2O$ and SNAPKV are executed in a chunked prefill manner.

| Phi-3-mini-128K on $\infty$Bench | | | | | |
|---|---|---|---|---|---|
| Method | R.Number | E.Sum | E.MC | C.Debug | Avg.$\uparrow$ |
| FULLATTN | 97.12 | 17.92 | 55.46 | 23.10 | 48.40 |
| $H_2O$ | 3.39 | 15.35 | 45.41 | 20.57 | 21.18 |
| SNAPKV | 2.54 | 15.44 | 41.92 | 21.43 | 20.33 |
| **Locret** | **97.46** | **16.82** | **46.29** | **29.71** | **47.57** |

**KV Cache and Chunked Prefill.** Given the input prompt sequence $\{t_1, \cdots, t_n\}$, during the prefill stage, all prompt tokens are processed in a single forward pass. After the prefill, $\mathbf{K}^{(i)} = \left[\mathbf{K}_1^{(i)}, \cdots, \mathbf{K}_{h/g}^{(i)}\right]$ and $\mathbf{Q}^{(i)} = \left[\mathbf{Q}_1^{(i)}, \cdots, \mathbf{Q}_{h/g}^{(i)}\right]$ are stored as the KV cache, whose sequence length is $n$. During the decoding stage, each time a token is decoded, a forward pass is conducted only for this token and decode the next token. In this process, the KV cache is used to avoid redundant attention computation. Chunked prefill is a method for reducing peak memory usage by segmenting input sequence and prefilling tokens chunk by chunk. Considering both the KV cache and chunked prefill, the attention block can be modified as:

$$\mathbf{A}[n+1\colon n+B]_j^{(i)} = \texttt{softmax}\left(\mathbf{M} \odot \frac{\mathbf{Q}[n+1\colon n+B]_j^{(i)}\mathbf{K}[1\colon n+B]_{\lceil j/g\rceil}^{(i)\top}}{\sqrt{d_m}}\right)\mathbf{V}[1\colon n+B]_{\lceil j/g\rceil}^{(i)}, \qquad (2)$$

where $\mathbf{A}[n+1\colon n+B]$ denotes the attention output for the tokens $\{n+1, \cdots, n+B\}$, and $B$ is the number of tokens processed in a single forward pass. For decoding, $B=1$, while for chunked prefill, $B$ corresponds to the chunk size. For the $k$-th token in the context, its attention output is $\mathbf{A}[k]$, its key and value vectors are $\mathbf{K}[k]$ and $\mathbf{V}[k]$.

**Cache Eviction.** In the cache eviction process, we treat the KV vector pair of a single token within one attention head as the smallest cache unit. We denote the cache unit of the $k$-th token as $c_k = (\mathbf{K}[k], \mathbf{V}[k])$. Assuming a memory budget $b$, representing the maximum number of cache units that can be stored, the abstract form of the attention block can be written as $c_k = f(c_1, c_2, \cdots, c_{k-1})$. With limited cache capacity, this process can only be approximated by $\tilde{c}_k = f(\tilde{c}_{p_1}, \tilde{c}_{p_2}, \cdots, \tilde{c}_{p_{b'}})$, where $b' \leq b$, and $p_1, p_2 \cdots, p_{b'} \in \{1, 2, \cdots, k-1\}$. When the cache is full, one unit must be evicted. We select the victim using some policy $p_v = \text{Policy}(\tilde{c}_{p_1}, \cdots, \tilde{c}_{p_b}; \tilde{c}_k)$, and the key challenge is to develop a policy that minimizes the error $\|\tilde{c}_k - c_k\|$.

### 3.2 The Global and Local Discrepancy of Scoring Functions in Existing Methods

Conducting KV cache eviction in conjunction with chunked prefill offers a potential solution for reducing peak GPU memory usage during long-context prefill; such a combination utilizes a scoring function of KV pair importance to identify essential KVs. Cache importance scoring functions can generally be categorized into two types: causal and non-causal. Existing causal functions, e.g., SIRLLM (Yao et al., 2024), predict cache importance without relying on future information, which, as shown in Section 5, leads to suboptimal performance. Non-causal functions require information from subsequent cache units to determine the importance score of a given cache unit, making them dependent on prefilling the entire sequence. Since their requirement for complete sequence information cannot be satisfied in the chunked prefill process, this introduces a significant discrepancy in importance estimation when based on global versus local context. In this section, we analyze two major KV cache eviction methods, $H_2O$ (Zhang et al., 2024d) and SNAPKV (Li et al., 2024b), to illustrate this discrepancy and its impact on task performance.

To quantify the global and local discrepancy, we define the *consistency* as follows. Given a sequence of $n$ KV pairs $c_1, c_2, \cdots, c_n$ and a scoring function $S(c_i|c_{j_1}, \cdots, c_{j_m})$ (representing the importance of KV pair $c_i$ with respect to $\{c_{j_1}, \cdots, c_{j_m}\}$), we calculate the global importance scores $s_1, s_2, \cdots, s_n$ and the local importance

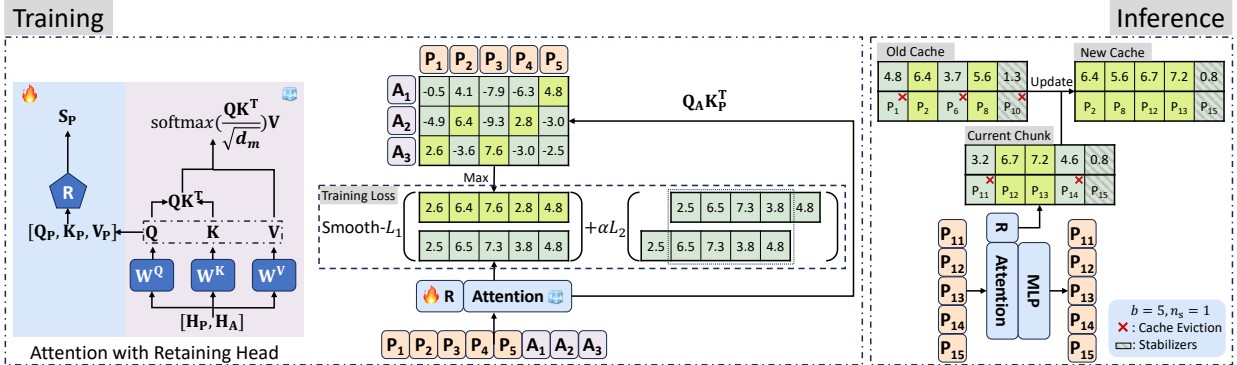

Figure 2: The framework of LOCRET. "**R**" represents the retaining head. $P_i$ and $A_i$ correspond to the $i$-th prompt token and answer token. "$b$" represents the budget size, and "$n_s$" represents the length of the stabilizers. For simplicity, our notation here does not reflect the concept of layers.

scores $s_1', s_2', \cdots, s_n'$ by

$$s_i = S(c_i|c_1, c_2, \cdots, c_n), \ \ s_i' = S(c_i|c_1, c_2, \cdots, c_i). \tag{3}$$

Then, the consistency of two scores are defined as

$$p_k = \frac{1}{k}|\text{argtop}_k(s_1, \cdots, s_n) \cap \text{argtop}_k(s_1', \cdots, s_n')| \in [0, 1]. \tag{4}$$

We show the the consistency of the top 10% essential KV pairs, i.e. $p_{0.1n}$, and the task performance evaluated on four representing tasks of $\infty$Bench (Zhang et al., 2024b) in Figure 1 and Table 1.

The results highlight that scoring functions requiring future information ($H_2O$ and SNAPKV) suffer from significant discrepancies when subsequent cache units are not considered. The top 10% essential KV identification of the first 0.5K tokens can only achieve less than 80% and 40% accuracy compared with considerating the complete sequence for $H_2O$ and SNAPKV correspondingly. Such discrepancy leads to the failure of $H_2O$ and SNAPKV in accurately retrieving information from the context, particularly in R.Number. Specifically, the model is unable to identify the importance of certain cache units at the time they are first encountered. Our proposed LOCRET, however, avoids such inconsistencies and achieves superior performance.

# 4 Locret : KV Cache Eviction with Causal Importance Score

## 4.1 Framework of Locret

LOCRET is a training-based KV cache eviction framework that works in conjunction with chunked prefill. As illustrated in Figure 2, LOCRET operates in two stages: training and inference. In the training stage, we modify the original LLM by appending a retaining head **R** to each attention module. We then train the retaining heads **R** while keeping the LLM backbone frozen. During the chunked prefill inference stage, the retaining heads **R** can obtain the importance of each cache unit. We retain the cache units with higher scores, along with stabilizers (i.e., the last tokens), in the cache pool located in GPU memory. Through this process, the retaining heads **R** learn and predict the patterns discovered by existing methods, e.g. attention sink in Xiao et al. (2024c) and vertical lines in MINFERENCE (Jiang et al., 2024a), as detailed in Appendix J.

The eviction policy assigns each cache unit an importance score reflecting its influence on comprehending subsequent context. This estimation is causal, termed the *causal importance score (CIS)*. The CIS of the $k$-th unit depends only on the preceding units and the $k$-th unit itself. Due to memory constraints, calculating the exact CIS on-chip is impractical. Intuitively, discarding certain KV cache during the attention calculation introduces errors, manifesting as biases in the CIS. However, as long as the importance estimation is sufficiently accurate, the core information can be preserved, and the loss of information in the attention process can be minimized. Therefore, the error introduced in the CIS can be neglected.

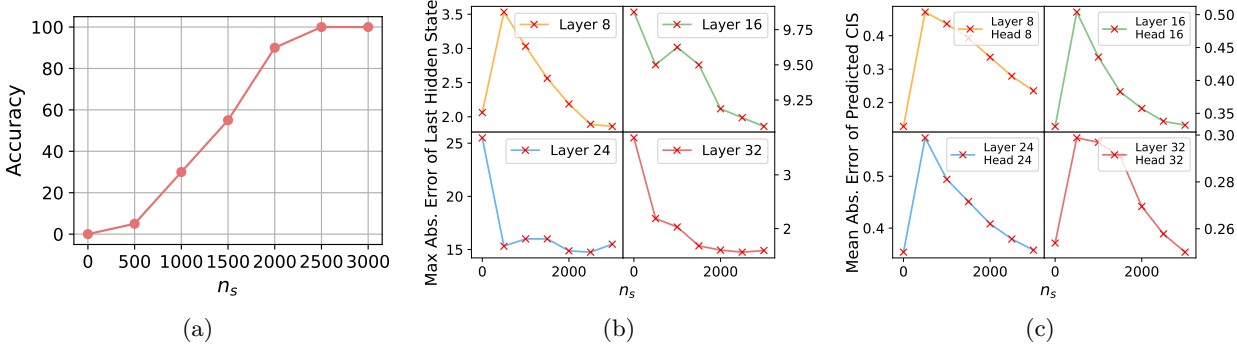

Figure 3: R.Number with different stabilizer lengths $n_s$. (a) Task accuracy under different $n_s$. (b) Maximum absolute error of the last hidden state. (c) Mean absolute error of the predicted CIS. We conduct this experiment on entries 101-120 of R.Number using the `Phi-3-mini-128K`.

## 4.2 Training Retaining Heads

In this section, we introduce LOCRET's model architecture modifications and the corresponding training recipe. We add additional parameters to compute the CIS $\mathbf{S}[k]$ for the $k$-th cache unit. Specifically, we inject a retaining head, consisting of a small MLP, into each layer. This design stems from the observation that a lightweight MLP can effectively learn essential KVs and minimize training loss, eliminating the need for more complex architectures. From our observation, such small MLPs do not slow down model inference, with details elaborated in Appendix I. The retaining head predicts the CIS for each head of the corresponding layer based on the concatenation of $[\mathbf{Q}, \mathbf{K}, \mathbf{V}]$. Formally, with a slight abuse of notation, let the retaining head for layer $i$ be denoted as $\mathbf{R}$. The CIS at head $j$ is then calculated as: $\tilde{\mathbf{S}} = \mathbf{R}([\mathbf{Q}, \mathbf{K}, \mathbf{V}]) = \sigma([\mathbf{Q}, \mathbf{K}, \mathbf{V}]\mathbf{W}_1)\mathbf{W}_2$. In this equation, $\mathbf{W}_1 \in \mathbb{R}^{(d_m + 2d_{kv}) \times d_\mathbf{R}}$ and $\mathbf{W}_2 \in \mathbb{R}^{d_\mathbf{R} \times \frac{h}{g}}$ are the tunable parameters of $\mathbf{R}$, and $\sigma$ is the activation function, $\tilde{\mathbf{S}}[k] = \left[\tilde{\mathbf{S}}[k]_1, \cdots, \tilde{\mathbf{S}}[k]_{h/g}\right] \in \mathbb{R}^{\frac{h}{g}}$, where $\tilde{\mathbf{S}}[k]_j$ is the predicted CIS of the $k$-th token. This architecture implies that the importance estimation for a single head is not performed in isolation but considers all heads.

We train the retaining head $\mathbf{R}$s on a small Question-Answer (QA) supervised fine-tuning (SFT) dataset, where each entry consists of a single prompt and one answer. We define the CIS $\mathbf{S}[k]_j$ for the $k$-th token at head $j$ as the maximum attention score, before softmax, from all the answer tokens toward the $k$-th token. Formally, given a training instance $d$, for the $k$-th token at head $j$ of layer $i$, we approximate the predicted value $\tilde{\mathbf{S}}[k]_j^{(i)}$ to the ground truth $\mathbf{S}[k]_j^{(i)} := \max_p \left(\mathbf{Q}_j^{(i)}\mathbf{K}_j^{(i)T}\right)_{p,k}$, where $n_q(d) < p \leq n_q(d) + n_a(d)$, and $n_q(d)$ and $n_a(d)$ represent the lengths of the prompt and answer in data $d$, respectively. For an MHA model with $L$ layers and $h$ heads, the training objective is described in Equation 5. For GQA models, we take the maximum attention score before softmax across different query heads within the same group as the ground truth for the corresponding KV head. This design leverages that semantically relevant tokens yield higher attention scores, while answer tokens (e.g., "the") are not. Thus, we use max pooling to filter them out.

$$\underset{\mathbf{W}_1^{(i)}, \mathbf{W}_2^{(i)}, i=1,2\cdots,L}{\operatorname{argmin}} \mathbb{E}_{d \in \mathcal{D}} \left[\sum_{i=1}^{L} \sum_{j=1}^{h} \sum_{k=1}^{n_q(d)} \mathcal{L}\left(\tilde{\mathbf{S}}[k]_j^{(i)}, \mathbf{S}[k]_j^{(i)}\right)\right] \tag{5}$$

The training loss consists of a regression loss and a smoothing loss. We apply the Smooth-$\mathcal{L}_1$ norm between the predicted values and the ground truth. Since important segments in natural language typically consist of adjacent tokens, we also apply the $\mathcal{L}_2$ norm between each pair of adjacent predicted values to enforce smoothness. The complete training loss for LOCRET is given by Equation 6.

$$\mathcal{L}\left(\tilde{\mathbf{S}}[k]_j^{(i)}, \mathbf{S}[k]_j^{(i)}\right) = \text{Smooth-}\mathcal{L}_1\left(\tilde{\mathbf{S}}[k]_j^{(i)}, \mathbf{S}[k]_j^{(i)}\right) + \alpha\mathcal{L}_2\left(\tilde{\mathbf{S}}[k]_j^{(i)}, \tilde{\mathbf{S}}[k+1]_j^{(i)}\right) \tag{6}$$

From our observations, the training of LOCRET exhibits strong robustness. Despite changes in $d_\mathbf{R}$ and the dataset, the performance variations shown in Figure 6 and Table 8 are minimal.

---

**Algorithm 1:** LOCRET Inference

---

**Input:** Model $\mathbf{M}$, Prompt tokens $x$, Local length $n_{loc}$, Stablizer length $n_s$, Budget $b$, Chunk size $B$
**Output:** Generated tokens $x_{gen}$
`// Leave the last `$n_{loc}$` out to make sure they are not evicted.`
chunk_positions ← split_chunk(0, $x$.length() $-n_{loc}$, $B$)
K_cache, V_cache, score_cache ← [], [], []
**for** chunk $\in$ chunk_positions **do**
    begin_pos, end_pos ← chunk.begin_pos, chunk.end_pos
    K_chunk, V_chunk, score_chunk ← $\mathbf{M}$($x$[begin_pos:end_pos], K_cache, V_cache)
    K_cache ← Concat(K_cache, K_chunk)
    V_cache ← Concat(V_cache, V_chunk)
    score_cache ← Concat(score_cache, score_chunk)
    **if** chunk is not the last chunk **then**
        `// Keep the last `$n_s$` caches to maintain higher context continuity.`
        score_cache[score_cache.length()-$n_s$:score_cache.length()] ← $+\infty$
    indices ← top-$b$(score_cache).indices
    K_cache, V_cache, score_cache = K_cache[indices], V_cache[indices], score_cache[indices]
K_cache, V_cache, score_cache ← $\mathbf{M}$($x$[$x$.length()$-n_{loc}$:$x$.length()], K_cache, V_cache)
$x_{gen}$ ← $\mathbf{M}$.generate(K_cache, V_cache)
**return** $x_{gen}$

---

### 4.3 Inference Implementation of Locret

During the inference stage, we use the chunked prefill pattern and perform cache eviction based on the predicted CIS. Since the predicted CIS does not rely on subsequent tokens, it remains consistent once calculated. Thus, we store the KV cache units along with their corresponding causal importance values. When the cache is full, we evict the units with lower causal importance values, as they are deemed less useful for future computations. Such eviction is performed during chunked prefill. When processing a new chunk, we first compute its KV cache, concatenate it with the previously retained cache, and evict redundant units to adhere to the budget size. Note that we cache the pre-RoPE KV cache and reassign continuous position embeddings from the beginning to enhance context continuity.

Cache eviction introduces context discontinuity, meaning some cache units at certain positions may be absent, which can degrade generation quality. To mitigate this, we retain the last $n_s$ tokens of the current chunk, named as the *stabilizers*, at each step of chunked prefill, ensuring a local and continuous context to minimize errors. As shown in Figure 3, smaller $n_s$ results in severe performance degradation, and the model fails entirely when stabilizers are absent, as context discontinuity leads to instability in CIS prediction, causing errors in cache eviction and amplifying errors in hidden states. More details are discussed in Appendix H. We provide a pseudocode of LOCRET inference in Algorithm 1.

### 4.4 Processing Query-Driven Tasks

Query-driven tasks are characterized by highly sparse yet query-correlated critical regions within the context. A representative example is the Multikey-NIAH task from RULER (Hsieh et al., 2024). Such contexts are inherently challenging to compress effectively without query information. To effectively handle such tasks, we introduce LOCRET-Q, a query-aware variant of LOCRET. When training the retaining heads, we prepend the last $l_q$ query tokens to the sequence and gather CIS labels, to avoid dismatch between training and inference. At inference, the query is inserted at the sequence start, ensuring its visibility across all eviction actions. This adaptation enables LOCRET-Q to perform *query-aware* eviction.

## 5 Experiments

We conduct experiments to evaluate whether LOCRET can address the following questions:

Table 2: Benchmark results on ∞Bench. "Avg." represents the average score across all tasks. The highest score in each column is marked in **bold**, and the second highest is underlined.

| Method | R.PassKey | R.Number | E.Sum | E.QA | E.MC | Z.QA | E.Dia | C.Debug | M.Find | Avg.↑ |
|---|---|---|---|---|---|---|---|---|---|---|
| | | | | Phi-3-mini-128K on ∞Bench | | | | | | |
| FULLATTN | $98.64_{\pm0.00}$ | $97.12_{\pm0.00}$ | $17.92_{\pm0.00}$ | $11.16_{\pm0.00}$ | $55.46_{\pm0.00}$ | $14.83_{\pm0.00}$ | $8.00_{\pm0.00}$ | $23.10_{\pm0.00}$ | $17.43_{\pm0.00}$ | $38.18_{\pm0.00}$ |
| INFLLM | $\mathbf{100.00}_{\pm0.00}$ | $\mathbf{97.12}_{\pm0.00}$ | $14.35_{\pm0.00}$ | $4.97_{\pm0.00}$ | $38.86_{\pm0.00}$ | $11.04_{\pm0.00}$ | $3.50_{\pm0.00}$ | $\underline{25.38}_{\pm0.00}$ | $15.14_{\pm0.00}$ | $\underline{34.48}_{\pm0.00}$ |
| HF-2BITS | $0.00_{\pm0.00}$ | $0.00_{\pm0.00}$ | $13.80_{\pm0.00}$ | $1.44_{\pm0.00}$ | $1.75_{\pm0.00}$ | $0.20_{\pm0.00}$ | $0.50_{\pm0.00}$ | $0.00_{\pm0.00}$ | $0.57_{\pm0.00}$ | $2.03_{\pm0.00}$ |
| SIRLLM | $3.39_{\pm0.00}$ | $3.39_{\pm0.00}$ | $\mathbf{21.06}_{\pm0.00}$ | $6.32_{\pm0.00}$ | $\mathbf{44.98}_{\pm0.00}$ | $\mathbf{11.99}_{\pm0.00}$ | $\underline{5.00}_{\pm0.00}$ | $22.34_{\pm0.00}$ | $\underline{21.71}_{\pm0.00}$ | $15.58_{\pm0.00}$ |
| MINFERENCE | $99.32_{\pm0.00}$ | $\underline{95.93}_{\pm0.00}$ | $14.44_{\pm0.00}$ | $\mathbf{8.11}_{\pm0.00}$ | $40.61_{\pm0.00}$ | $10.60_{\pm0.00}$ | $\mathbf{9.00}_{\pm0.00}$ | $15.48_{\pm0.00}$ | $15.43_{\pm0.00}$ | $34.32_{\pm0.00}$ |
| **Locret** | $\underline{99.88}_{\pm0.10}$ | $91.30_{\pm7.55}$ | $\underline{16.75}_{\pm0.85}$ | $\underline{8.03}_{\pm0.38}$ | $\underline{43.38}_{\pm2.52}$ | $\underline{11.11}_{\pm0.21}$ | $\mathbf{9.00}_{\pm1.73}$ | $\mathbf{26.65}_{\pm2.42}$ | $\mathbf{30.19}_{\pm1.35}$ | $\mathbf{37.37}_{\pm1.14}$ |
| | | | | Llama-3.1-8B-instruct on ∞Bench | | | | | | |
| FULLATTN | $100.00_{\pm0.00}$ | $99.32_{\pm0.00}$ | $26.79_{\pm0.00}$ | $15.06_{\pm0.00}$ | $68.12_{\pm0.00}$ | $13.40_{\pm0.00}$ | $17.00_{\pm0.00}$ | $20.56_{\pm0.00}$ | $34.00_{\pm0.00}$ | $43.81_{\pm0.00}$ |
| INFLLM | $\mathbf{100.00}_{\pm0.00}$ | $\mathbf{100.00}_{\pm0.00}$ | $24.24_{\pm0.00}$ | $14.21_{\pm0.00}$ | $51.97_{\pm0.00}$ | $10.76_{\pm0.00}$ | $11.00_{\pm0.00}$ | $\underline{26.25}_{\pm0.00}$ | $\mathbf{35.71}_{\pm0.00}$ | $41.57_{\pm0.00}$ |
| HF-2BITS | $36.78_{\pm0.00}$ | $6.95_{\pm0.00}$ | $8.77_{\pm0.00}$ | $4.05_{\pm0.00}$ | $27.95_{\pm0.00}$ | $3.09_{\pm0.00}$ | $5.50_{\pm0.00}$ | $13.20_{\pm0.00}$ | $22.00_{\pm0.00}$ | $14.25_{\pm0.00}$ |
| SIRLLM | $1.69_{\pm0.00}$ | $1.69_{\pm0.00}$ | $\underline{25.60}_{\pm0.00}$ | $8.95_{\pm0.00}$ | $55.46_{\pm0.00}$ | $10.38_{\pm0.00}$ | $9.50_{\pm0.00}$ | $23.10_{\pm0.00}$ | $3.71_{\pm0.00}$ | $15.56_{\pm0.00}$ |
| MINFERENCE | $\mathbf{100.00}_{\pm0.00}$ | $98.47_{\pm0.00}$ | $20.64_{\pm0.00}$ | $\underline{14.35}_{\pm0.00}$ | $\mathbf{59.83}_{\pm0.00}$ | $\mathbf{12.20}_{\pm0.00}$ | $\mathbf{20.50}_{\pm0.00}$ | $25.89_{\pm0.00}$ | $\underline{35.43}_{\pm0.00}$ | $\underline{43.03}_{\pm0.00}$ |
| **Locret** | **100.00** | $\underline{99.49}$ | **27.28** | **20.90** | $\underline{58.82}$ | $\underline{11.85}$ | $\underline{13.00}$ | **27.16** | 32.86 | **43.48** |

Table 3: Benchmark results on L-Eval. "Avg." represents the average score across all tasks. The highest score in each column is marked in **bold**, and the second highest is underlined.

| Method | CodeU | NQ | CUAD | NarrativeQA | QMSum | SPACE | Avg.↑ |
|---|---|---|---|---|---|---|---|
| | | | Phi-3-mini-128K on L-Eval | | | | |
| FULLATTN | $8.89_{\pm0.00}$ | $59.14_{\pm0.00}$ | $30.34_{\pm0.00}$ | $17.59_{\pm0.00}$ | $16.05_{\pm0.00}$ | $14.51_{\pm0.00}$ | $24.42_{\pm0.00}$ |
| INFLLM | $5.56_{\pm0.00}$ | $34.32_{\pm0.00}$ | $14.53_{\pm0.00}$ | $14.80_{\pm0.00}$ | $13.31_{\pm0.00}$ | $\underline{14.81}_{\pm0.00}$ | $16.22_{\pm0.00}$ |
| HF-2BITS | $0.00_{\pm0.00}$ | $1.69_{\pm0.00}$ | $6.40_{\pm0.00}$ | $2.04_{\pm0.00}$ | $2.73_{\pm0.00}$ | $3.34_{\pm0.00}$ | $2.70_{\pm0.00}$ |
| SIRLLM | $\mathbf{8.89}_{\pm0.00}$ | $\underline{37.92}_{\pm0.00}$ | $20.89_{\pm0.00}$ | $14.51_{\pm0.00}$ | $13.70_{\pm0.00}$ | $14.46_{\pm0.00}$ | $\underline{18.40}_{\pm0.00}$ |
| MINFERENCE | $7.78_{\pm0.00}$ | $25.21_{\pm0.00}$ | $\mathbf{26.64}_{\pm0.00}$ | $\mathbf{15.14}_{\pm0.00}$ | $\mathbf{15.78}_{\pm0.00}$ | $14.87_{\pm0.00}$ | $17.57_{\pm0.00}$ |
| **Locret** | $\underline{7.78}_{\pm1.92}$ | $\mathbf{52.37}_{\pm0.95}$ | $\underline{22.34}_{\pm0.33}$ | $\underline{15.01}_{\pm0.14}$ | $\underline{15.53}_{\pm0.79}$ | $13.88_{\pm0.16}$ | $\mathbf{21.15}_{\pm0.26}$ |
| | | | Llama-3.1-8B-instruct on L-Eval | | | | |
| FULLATTN | $10.00_{\pm0.00}$ | $66.84_{\pm0.00}$ | $38.91_{\pm0.00}$ | $23.11_{\pm0.00}$ | $18.76_{\pm0.00}$ | $16.86_{\pm0.00}$ | $29.08_{\pm0.00}$ |
| INFLLM | $6.67_{\pm0.00}$ | $54.77_{\pm0.00}$ | $33.76_{\pm0.00}$ | $20.35_{\pm0.00}$ | $17.62_{\pm0.00}$ | $16.73_{\pm0.00}$ | $24.98_{\pm0.00}$ |
| HF-2BITS | $1.11_{\pm0.00}$ | $29.79_{\pm0.00}$ | $18.98_{\pm0.00}$ | $9.46_{\pm0.00}$ | $14.02_{\pm0.00}$ | $13.73_{\pm0.00}$ | $14.52_{\pm0.00}$ |
| SIRLLM | $5.56_{\pm0.00}$ | $\underline{58.00}_{\pm0.00}$ | $35.41_{\pm0.00}$ | $\underline{21.21}_{\pm0.00}$ | $17.32_{\pm0.00}$ | $16.44_{\pm0.00}$ | $\underline{25.66}_{\pm0.00}$ |
| MINFERENCE | $\underline{7.78}_{\pm0.00}$ | $31.80_{\pm0.00}$ | $\underline{36.93}_{\pm0.00}$ | $19.44_{\pm0.00}$ | $\underline{18.14}_{\pm0.00}$ | $\underline{16.76}_{\pm0.00}$ | $21.81_{\pm0.00}$ |
| **Locret** | $\underline{7.78}_{\pm1.78}$ | $\mathbf{63.90}_{\pm0.94}$ | $\mathbf{37.20}_{\pm0.11}$ | $\mathbf{23.54}_{\pm0.07}$ | $\mathbf{18.54}_{\pm0.32}$ | $\mathbf{16.94}_{\pm0.07}$ | $\mathbf{27.99}_{\pm0.38}$ |

(**Q1**) Can LOCRET obtain better end-to-end task performance compared to popular and competitive long-context inference methods using similar or less peak memory?

(**Q2**) Is LOCRET able to achieve a faster inference speed on consumer-grade devices?

(**Q3**) How can LOCRET process query-driven tasks?

(**Q4**) Can LOCRET achieve stable performance across various hyperparameter settings and training recipes?

## 5.1 Experimental Setup

**Models and Datasets.** We conduct experiments on two long-context LLMs: Phi-3-mini-128K (Abdin et al., 2024) and Llama-3.1-8B-instruct (Dubey et al., 2024). Both models can process up to 128K context tokens and follow MHA and GQA architectures, respectively. The parameter sizes of these two models are also suitable for deployment on consumer-grade devices. We inject retaining heads **R** into each layer of these two models, and the intermediate size $d_{\mathbf{R}}$ is 1,024. The retaining heads are trained on the LongAlpaca dataset (Chen et al., 2024) for 3,000 steps , with a 5e-4 learning rate, 10,240 sequence length, and $\alpha$ set to 2.5e-3. Training LOCRET is lightweight, with the tunable parameters comprising 8% and 2.5% of the total for the two models, respectively. The complete training process takes 0.47 and 0.80 GPU hours on an A800 GPU for each corresponding model. More important hyperparameters are listed in Table 9. More details on hyperparameters and system environments can be found in Appendix A.

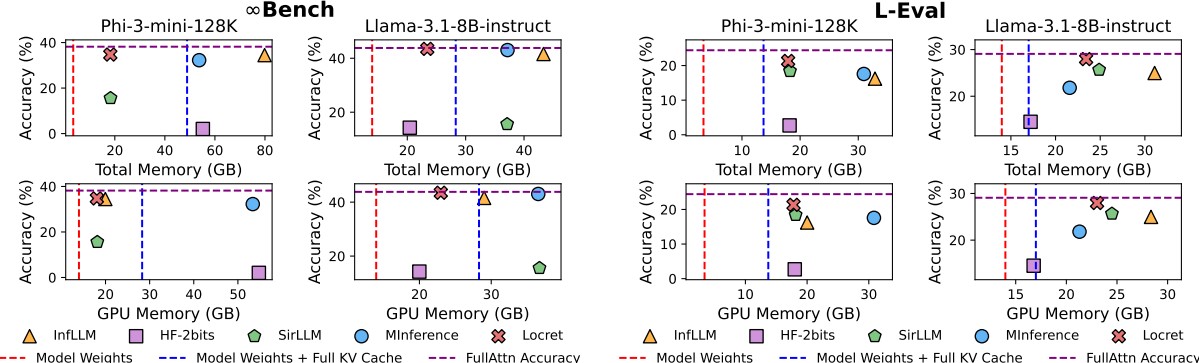

Figure 4: Memory Statistics vs. Task Performance. The red lines correspond to the theoretical size of the model weights, while the blue lines represent the total size of the model weights and the full KV cache without any compression. The purple lines indicate the accuracies of FULLATTN. "Total Memory" represents the total memory usage of both GPU and CPU.

**Benchmarks.** We evaluate LOCRET on selected subsets of ∞Bench (Zhang et al., 2024b) and L-Eval (An et al., 2024). For query-driven tasks, we evaluate LOCRET on RULER (Hsieh et al., 2024). For ∞Bench, we select R.PassKey, R.Number, E.Sum, E.QA, E.MC, Z.QA, E.Dia, C.Debug, and M.Find. All selected subsets, except Z.QA, have an average length of approximately 100K tokens, while Z.QA has a longer average length of around 2000K tokens. We exclude R.KV because it can be easily handled by calling a Python interpreter. We also exclude C.Run and M.Calc due to their complexity for all methods, including full attention inference. For L-Eval, we filter out all tasks with an average length shorter than 16K tokens and evaluate models on CodeU, NQ, CUAD, NarrativeQA, QMSum, and SPACE. Metrics are reported according to the recommendations of the two datasets, with further details provided in Appendix A. Although temperature sampling can be useful in real-world applications, we adopt greedy generation aligned with baseline methods. Since LOCRET does not modify sampling, we believe this setup is representative. We also report the peak memory usage, i.e. the average peak memory measured for the first entry of each task in ∞Bench and L-Eval, for reference.

**Baselines.** As discussed in Section 2, existing algorithms for memory-efficient long-context inference can be categorized into offloading-based, sparsity-based, quantization-based, and token-dropping methods. For each category, we select one representative method as the baseline. We compare LOCRET against full attention inference (denoted as FULLATTN), INFLLM (Xiao et al., 2024a), MINFERENCE (Jiang et al., 2024a), KV cache quantization (Turganbay, 2024), and SIRLLM (Yao et al., 2024). For quantization, we use HuggingFace Quanto (Hugging-Face) implementation, referring to the 2-bit quantization method as HF-2BITS. We omit HF-4BITS and benchmark the combination with LOCRET in Appendix E. We do not include attention pooling-based methods in this benchmark, as they are orthogonal to our approach, and further discussion about this is provided in Appendix E. When conducting query-driven tasks, we compare LOCRET and LOCRET-Q with $H_2O$ (Zhang et al., 2024d), SNAPKV (Li et al., 2024b), SIRLLM, and INFINIPOT (Kim et al., 2024). Environmental setup and details of the selected baselines can be found in Appendix A.

### 5.2 End-to-end Benchmark

We compare all the methods on ∞Bench and L-Eval to address **Q1**. In Table 2 and Table 3, LOCRET outperforms baselines in terms of end-to-end performance, showing: (1) On ∞Bench, while all methods experience performance degradation compared to FULLATTN, LOCRET, INFLLM, and MINFERENCE exhibit better performance than other methods, with only a modest drop in performance given the reduced memory usage. Quantization shows significant degradation and fails on all tasks. SIRLLM performs well on comprehensive tasks such as E.Sum and E.MC, but struggles with tasks that require precise memory, such as R.PassKey and R.Number. LOCRET not only excels in context retrieval tasks but also achieves strong results in comprehensive tasks, earning the highest overall score among all competitors. (2) On L-Eval, all methods show performance degradation. Nevertheless, LOCRET achieves the best overall performance, obtaining the highest scores on most tasks. L-Eval is a shorter but more complex dataset, where SIRLLM performs particularly well. Quantization fails on most tasks. Both INFLLM and MINFERENCE suffer significant performance drops

Table 4: Executing R.PassKey on an NVIDIA 4090. "tok/s" represents the inference speed, "C.Len" stands for the context length after truncation, and "Acc." represents task accuracy. The highest score among 128K context is marked in **bold**.

| Method | | FullAttn | InfLLM | HF-2bits | SirLLM | MInference | **Locret** | HF-2bits* | MInference* |
|---|---|---|---|---|---|---|---|---|---|
| Phi-3-mini-128K | tok/s↑ | - | 2276.38 | - | 2352.20 | - | **5080.85** | 1098.51 | 4099.92 |
| | C.Len.↑ | 128K | 128K | 128K | 128K | 128K | **128K** | 30K | 14K |
| | Acc.↑ | OOM | 99.83 | OOM | 1.69 | OOM | **100.00** | 0.00 | 13.56 |
| Llama-3.1-8B-instruct | tok/s↑ | - | 2287.66 | 1365.51 | 1589.75 | - | **3209.10** | 3680.06 | 5135.74 |
| | C.Len.↑ | 128K | 128K | 128K | 128K | 128K | **128K** | 30K | 25K |
| | Acc.↑ | OOM | 100.00 | 35.59 | 1.69 | OOM | **100.00** | 26.78 | 20.34 |

compared to FullAttn inference. Locret consistently surpasses all competitors. Notably, Locret can even surpass FullAttn on certain tasks, such as R.Number and C.Debug in ∞Bench. This is because the critical regions in the input context that contain the correct answers can be distinguished from the surrounding noise. Evicting irrelevant tokens reduces input noise, thereby enhancing the model's ability to retrieve the correct information. We believe this phenomenon highlights a beneficial property for long-context processing.

We report memory consumption in Figure 4, showing: In the extreme long-context scenario (∞Bench), Locret uses relatively less memory while achieving the best overall performance. InfLLM performs well with limited GPU memory usage, but it requires a significant amount of CPU memory to store the full KV cache. HF-2Bits and SirLLM can achieve low memory consumption in some settings, but quantization introduces severe performance degradation. MInference employs sparse attention patterns but does not compress the KV cache. As a result, its minimum memory requirement equals the sum of the model weights and the full KV cache. In the shorter-context scenario (L-Eval), memory usage is mainly determined by the KV-cache size. For `Phi-3-mini-128K`, which employs MHA and thus has a larger cache, MInference still consumes substantial GPU memory due to the lack of KV reduction, while InfLLM lowers GPU memory by offloading at the cost of higher total memory. SirLLM achieves memory usage comparable to Locret, and HF-2bits consumes similar memory but suffers severe performance degradation. For `Llama-3.1-8B-instruct`, whose KV cache is smaller, the bottleneck shifts to runtime memory for attention and other computations, so MInference and HF-2bits exhibit notably lower memory usage but significant performance loss. Among baselines with similar memory usage, Locret performs the best.

In summary, our experiments demonstrate that Locret is both effective and efficient, outperforming all baselines on multiple datasets and models while using less GPU memory.

### 5.3 Processing Speed on Real Consumer-Grade Devices

We examine the processing speed to demonstrate that Locret achieves its strong performance without compromising inference speed, addressing question **Q2**. We evaluate the inference speed on the R.PassKey task from ∞Bench and compare Locret against all the baselines, using a single NVIDIA 4090 GPU with 24GB of memory, which is typical for consumer-grade AI devices. We report the inference speed (the total number of tokens within the input and output sequences divided by the processing time) and the task accuracy. Since the default settings of some baselines can cause out-of-memory (OOM) errors, we simulate a common fallback used in GPU-constrained scenarios: truncating the long input from the middle. This approach incurs substantial performance degradation due to significant information loss, which also reflects real-world behavior when applied in practice. We mark these settings with * and report the corresponding valid context length. For settings without *, we maximize the chunk size for a higher inference speed.

R.PassKey is a task where the model retrieves a 5-digit number from a large amount of irrelevant text, a task we believe to be relatively simple for humans. Thus, we consider the task to have failed if the accuracy falls below 95%. As shown in Table 4, aside from the settings that fail on this task, Locret achieves the highest inference speed among all methods that can correctly process R.PassKey: (1) Due to its MHA architecture, `Phi-3-mini-128K` has a larger KV cache, which leads to failures for both HF-2bits and MInference. Storing the full KV cache on a single 4090 GPU is infeasible, as it requires 48GB of memory. Although the quantized KV cache is reduced to 6GB, converting representations requires significant GPU memory for

Table 5: Performance, prefill speed, and decode speed on RULER-128K. The best and second-highest scores among eviction-based methods in each column are highlighted in **bold** and underlined, respectively. We randomly generate 20 entries for each task for all settings. Prefill and decode speed are reported in tok/s. FULLATTN is implemented using FLASHATTN.

| Method | SG1 | SG2 | SG3 | MK1 | MK2 | MK3 | MV | MQ | VT | CWE | FWE | QA1 | QA2 | Avg. | Prefill | Decode |
|---|---|---|---|---|---|---|---|---|---|---|---|---|---|---|---|---|
| FULLATTN | 100.00 | 100.00 | 100.00 | 95.00 | 95.00 | 75.00 | 91.25 | 100.00 | 60.00 | 72.00 | 66.67 | 75.00 | 40.00 | 82.30 | 4319.95 | 12.32 |
| MINFERENCE | 100.00 | 95.00 | 100.00 | 95.00 | 15.00 | 10.00 | 81.25 | 93.75 | 77.00 | 14.50 | 70.00 | 40.00 | 30.00 | 63.19 | 7205.06 | 2.61 |
| SNAPKV | **100.00** | 95.00 | 0.00 | 65.00 | 40.00 | 10.00 | 62.50 | **90.00** | 58.00 | 6.00 | 46.67 | 50.00 | 30.00 | 50.24 | 4203.34 | 36.49 |
| H2O | 20.00 | 0.00 | 0.00 | 0.00 | 0.00 | 0.00 | 3.75 | 2.50 | 1.00 | 0.00 | 53.33 | **100.00** | 15.00 | 15.04 | 464.73 | **44.70** |
| SIRLLM | 0.00 | 5.00 | 5.00 | 0.00 | 5.00 | 5.00 | 6.25 | 8.75 | 12.00 | 0.00 | **80.00** | 30.00 | 15.00 | 13.23 | **9717.41** | 40.86 |
| INFINIPOT | **100.00** | 40.00 | 0.00 | 15.00 | 0.00 | 0.00 | 15.00 | 31.25 | **93.00** | 26.50 | 51.67 | 20.00 | 15.00 | 31.34 | 9099.02 | 34.36 |
| LOCRET | **100.00** | 45.00 | 35.00 | 10.00 | 5.00 | 0.00 | 20.00 | 17.50 | 69.00 | **46.50** | 73.33 | 20.00 | 5.00 | 34.33 | 9587.09 | 37.38 |
| LOCRET-Q | **100.00** | **100.00** | **100.00** | **70.00** | **100.00** | **100.00** | **70.00** | 85.00 | 60.00 | 34.50 | **80.00** | 50.00 | **35.00** | **75.73** | 9587.84 | 40.11 |

its intermediate states, resulting in the failure of HF-2BITS. While INFLLM can run in memory-limited scenarios, its offloading process slows down inference. SIRLLM fails due to its inaccurate eviction, which cannot correctly identify the 5-digit number. (2) In the GQA model (`Llama-3.1-8B-instruct`), which has a smaller KV cache, the quantized cache can fit within the GPU memory. However, this method introduces no sparsity in attention computation or memory access, resulting in relatively lower inference speed compared with other methods. The performance of INFLLM, SIRLLM, and MINFERENCE is similar to that seen with `Phi-3-mini-128K`. Although MINFERENCE benefits from faster encoding speed, it fails this task because it cannot process the entire input sequence at once. LOCRET strikes a balance between inference speed and performance, making it a far more suitable solution for long-context scenarios on consumer-grade devices.

## 5.4 Locret-Q: Supporting Query-Driven Tasks

We mainly address question **Q3** in this section. As introduced by Sun et al. (2024), existing eviction-based techniques exhibit significant performance degradation when applied to query-driven tasks. Due to this, we evaluate LOCRET against classical *eviction-based baselines* using the RULER benchmark (Hsieh et al., 2024).

Table 5 shows LOCRET-Q against SNAPKV (Li et al., 2024b), H2O (Zhang et al., 2024d), SIRLLM (Yao et al., 2024), and INFINIPOT (Kim et al., 2024) on RULER with a 128K context length. For reference, we also include results for FULLATTN and MINFERENCE. Metrics include prefill speed, decode speed, and task performances are reported. We randomly generate 20 entries for each task for all methods. As shown in Table 5, all efficient inference methods exhibit performance degradation to some extent. However, LOCRET-Q outperforms other eviction-based methods and even surpasses MINFERENCE, demonstrating its effectiveness on query-driven tasks. SNAPKV shows performance degradation, while H2O and SIRLLM fail completely on RULER. To further compare methods that do not fail completely, we conduct a higher-precision RULER benchmark (Table 7), generating 500 random entries per task. LOCRET-Q consistently outperforms SNAPKV. For speedup, methods combining eviction and chunked prefill (LOCRET-Q, LOCRET, SIRLLM, and INFINIPOT) significantly reduce prefill time, achieving > 2× speedup over FULLATTN. SNAPKV cannot accelerate prefill due to no computation reduction. H2O suffers from extremely slow prefill as it relies on full-sequence attention scores, incompatible with efficient implementations like FLASH-ATTN. Decoding speeds are similar across eviction-based methods, and all of them are faster than FULLATTN. Notably, LOCRET fails on RULER, showing a gap compared to LOCRET-Q, highlighting the necessity of query-awareness for query-centric tasks. A simple modification to LOCRET unlocks its potential for such tasks. To ablate the effect of placing the query at the beginning of the input sequence, we conduct an ablation study presented in Table 6. The results show that not all methods benefit from this modification; in contrast, LOCRET benefits the most, significantly outperforming the others. This indicates that the superior performance of LOCRET-Q is not due to early access to the query, but rather stems from its query-aware KV cache eviction mechanism.

## 5.5 Hyperparameter and Training Robustness Analysis

To adderss **Q4**, we first examine the inference hyperparameters: budget, stabilizer length, and chunk size.

Table 6: Ablation on putting the query in the front of the input. We randomly generate 20 entries for each task for all settings. The average accuracy of RULER-128K is reported.

Table 7: RULER-128K of 500 entries for each task.

| Query-Front | FullAttn | MInference | SnapKV | H$_2$O | SirLLM | InfiniPot | Locret |
|---|---|---|---|---|---|---|---|
| ✗ | 82.30 | 63.19 | 50.24 | 15.04 | 13.23 | 31.34 | 34.33 |
| ✓ | 75.66 | 54.19 | 48.09 | 8.89 | 13.09 | 32.47 | 75.73 |
| Δ(✓ - ✗) | -6.64 | -9.00 | -2.15 | -6.15 | -0.14 | 1.13 | **41.40** |

| Method | RULER-128K |
|---|---|
| FullAttn | 82.20 |
| MInference | 72.97 |
| SnapKV | 48.76 |
| Locret-Q | **75.54** |

**Budget.** To evaluate the robustness of Locret under different budget constraints, we compare the proposed method with SnapKV (Li et al., 2024b) using chunked prefill on LongBench (Bai et al., 2024b). As shown in Figure 5a, when the budget size increases, Locret demonstrates a faster performance improvement compared to SnapKV, and consistently outperforms SnapKV.

**Stabilizers Length.** As discussed in Figure 3, stabilizers play a crucial role in context retrieval tasks. However, in NLU tasks, the stability of $n_s$ remains relatively high. We evaluate QMSum with different stabilizer lengths $n_s$, with the budget set at 6000. As illustrated in Figure 5b, performance remains consistent when $n_s$ is small. The observed performance degradation at larger $n_s$ values is due to the reduced space available for other cache units.

**Chunk Size.** Executing long-context inference on hardware with varying GPU memory limitations choices of chunk size. When the chunk size changes, Locret shows stable performance. We test on the NQ dataset from L-Eval using multiple chunk sizes ranging from 256 to 4096. The results, shown in Figure 5c, highlight the stability under various chunk sizes.

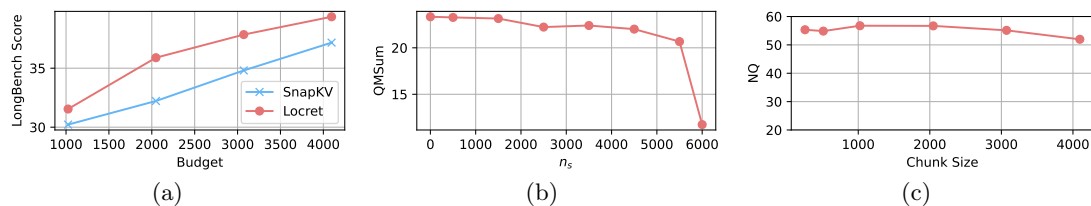

(a)          (b)          (c)

Figure 5: Scores of Locret under (a) various budgets; (b) various $n_s$; (c) various chunk size.

We further investigate Locret's training stability by experimenting with various intermediate sizes $d_{\mathbf{R}}$ of the retaining head and conducting training under different datasets. Locret demonstrates high robustness to these variations in both intermediate size and training recipes, indicating that it does not require careful hyperparameter tuning or meticulous dataset selection.

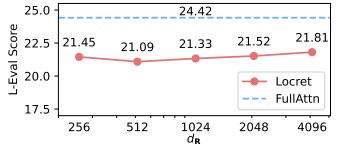

Figure 6: L-Eval scores with different $d_{\mathbf{R}}$.

Table 8: L-Eval scores of Locret trained on various datasets.

| Dataset | LongAlpaca | LongAlign | Anti-Haystack |
|---|---|---|---|
| L-Eval | 21.33 | 22.00 | 20.72 |

### 5.6 Additional Experiments

Additional experiments are included in the appendices. We evaluate Locret on LongBench in Appendix B, explore the combination of Locret with other methods in Appendix E, challenge Locret with extremely long inputs in Appendix F, and benchmark Locret in multi-turn conversation scenario in Appendix G.

## 6 Conclusion

We propose Locret, a lightweight training-based method that enables memory-efficient long-context LLM inference on consumer-grade devices. Locret introduces retaining heads to predict the CIS of each cache unit during chunked prefill and performs accurate cache eviction. We conduct extensive experiments across

different models and multiple datasets to compare LOCRET with major efficient inference techniques, and results show that LOCRET outperforms all baselines, using less GPU memory and without requiring offloading to CPU memory. LOCRET-Q, a query-aware variant of LOCRET, can further process query-centric tasks without significant performance degradation. Future work will involve testing LOCRET on other model architectures, e.g. encoder-decoders, and evaluating LOCRET on more personal devices, e.g. NVIDIA Jetson.

## Acknowledgment

This work is supported by the HKUST startup grant R9895 from CSE,RGC-ECS project 26218024, RGC-NSFC project CRS_HKUST601/24 and the high-quality development project of MIIT. Yuxiang Huang is supported by Beijing National Science Foundation (No. QY24253) and Tsinghua University Initiative Scientific Research Program (Student Academic Research Advancement Program). We acknowledge the discussion with Ruisi Cai for LoCoCo implementation, Xinrong Zhang for details of ∞Bench, Weilin Zhao for implementation issues, Chenyang Song for model sparsity, Shuo Wang for long-context training recipe and Yuan Yao for future works on multimodal LLMs. We also thank Minsoo Kim for discussions on InfiniPot.

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

# A    Hyperparameters, Environment and Baselines

## A.1    Training

During the training stage, we first insert retaining head $\mathbf{R}$s to each layar. A retaining head is a small FFN consist of two linear transformations, and the non-linear function is aligned with other non-linears of the corresponding model, with an intermediate size of 1024. We train the appended retaining head $\mathbf{R}$s on the LongAlpaca for 3000 steps with batch size set to 1 and maximum sequence length set to 10240. We use the AdamW scheduler (Loshchilov, 2019) and the learning rate is set to 5e-4. We conduct the training with a linear learning rate scheduler, whose warmup step number is set to 2000. The balance factor between two training loss $\alpha$ is set to 0.0025.

Table 9: Hyperparameters in LOCRET's inference stage. "$b$" is cache budget, "$B$" refers to chunk size of chunked prefill, "$n_s$" refers to stabilizers length and "$n_{loc}$" is local length.

| Model | $b$ | $B$ | $n_s$ | $n_{loc}$ |
|---|---|---|---|---|
| Phi-3-mini-128K | 6000 | 3072 | 2500 | 100 |
| Llama-3.1-8B-instruct | 16384 | 1024 | 2500 | 100 |

## A.2 Inference

The inference hyperparameters of LOCRET is listed in Table 9. Here, we follow the notations in Algorithm 1. $b$ stands for the cache budget, $B$ is the chunk size of chunked prefill, $n_s$ is the length of stabilizers, and $n_{loc}$ represents the length of locally retained tokens at the end of the input sequence.

Hyperparameters of other baselines are as follows. For INFLLM, we use the recommended settings for Llama-3 to evaluate Llama-3.1. Since there is no recommendations of Phi-3-mini-128K, we use the settings for MiniCPM, whose architechture and size is similar to Phi-3-mini-128K, to conduct all the experiments. For Quantization, we use the official implementation (Quanto backend) of Hugging Face. For SIRLLM, we set the start size to 4, recent size to 1000 for both models. We set the token entropy size to 6000 and 16384 for Phi-3-mini-128K and Llama-3.1-8B-instruct respectively. The chunk size of chunked prefill is also 3072 and 1024 for the corresponding model. For MINFERENCE, we utilize the recommended settings for both models.

## A.3 System Environment

For all the experiments except the 4090 experiments in Section 5.3, we use a workstation with 8×NVIDIA A800/H800 GPUs and 104 Intel(R) Xeon(R) Platinum 8470 CPUs. We only use 1 GPU from the cluster for training, as the GPU requirements are less than 80GB for all training procedures. The device has 1.0 TB CPU memory. The operating system is Red Hat 4.8.5. We conduct all experiments except the full attention full KV cache inference on a single GPU, and 2 GPUs for full attention settings.

For Section 5.3, we conduct the experiments on a single NVIDIA 4090 GPU. The device has 512 AMD EPYC 9754 128-Core Processors and 1.0 TB CPU memory. GPUs and CPUs are connected through PCIe Gen 4, which has 16GT/s transmission speed. The operating system is Ubuntu 9.4.0.

## A.4 Baselines

We compare LOCRET with full attention inference, INFLLM, Quantization, SIRLLM and MINFERENCE. FULLATTN inference is performed using vllm (Kwon et al., 2023), which includes automatic tensor parallelism. INFLLM is a representative of the offloading-based methods, where the full KV cache is offloaded to CPU, and the most relavant blocks are retrieved to GPU during inference. For quantization method, we use the Hugging Face implementation of 2-bits KV cache quantization, which is inspired by Liu et al. (2024b), where quantization is conducted along channels instead of tokens. We denote this method as HF-2BITS. SIRLLM is an eviction-based token dropping algorithm, where tokens with low token-entropy is evicted once the cache is fullfilled. We use the official implementation of SirLLM, which includes some CPU operations including importance sorting. MINFERENCE is a typical method of reducing peak GPU memory consumption through rule-based sparse attention, but it does not reduce the size of KV cache. Note that INFLLM, HF-2BITS and SIRLLM does not have official implementation on Phi-3-mini-128K, thus we implement these three methods according to the original algorithm. We only use the short factor of RoPE for INFLLM, and no further model modification is conducted for HF-2BITS and SIRLLM.

Table 10: LongBench scores of LOCRET compared with baselines.

| Method | gov_ report | triviaqa | narrative qa | qmsum | musique | 2wikimqa | multifield qa_en | repobench -p |
|---|---|---|---|---|---|---|---|---|
| FULLATTN | 33.35 | 86.38 | 18.21 | 19.51 | 19.82 | 33.37 | 49.82 | 58.02 |
| MINFERENCE | 32.94 | **86.87** | 19.46 | 19.57 | 18.85 | 33.30 | 49.14 | 58.98 |
| SIRLLM | 32.92 | 85.61 | 21.08 | 21.59 | 24.32 | 34.97 | 48.52 | **59.15** |
| INFLLM | 25.96 | 84.87 | 20.83 | 19.61 | 13.63 | 27.43 | 41.29 | 55.73 |
| **Locret** | **33.46** | 82.39 | **24.56** | **23.35** | **25.12** | **35.93** | **52.77** | 57.16 |

| Method | qasper | hotpotqa | multi_ news | trec | passage_ retrieval_en | passage _count | samsum | lcc | Avg.↑ |
|---|---|---|---|---|---|---|---|---|---|
| FULLATTN | 41.07 | 43.06 | **26.57** | 67.00 | 93.50 | 2.97 | 23.15 | 51.86 | 41.73 |
| MINFERENCE | **40.31** | 43.56 | 26.35 | **68.00** | **89.00** | 2.10 | 25.58 | 53.68 | 41.73 |
| SIRLLM | 40.17 | 47.00 | 26.44 | 65.50 | 63.00 | 3.00 | 23.11 | 51.83 | 40.51 |
| INFLLM | 30.51 | 38.05 | 25.36 | 64.50 | 10.00 | **7.50** | 0.28 | **61.59** | 32.95 |
| **Locret** | 40.17 | **48.70** | 26.41 | 62.00 | 83.00 | 3.00 | **26.37** | 52.61 | **42.31** |

## A.5 RULER Benchmark

To evaluate LOCRET-Q's performance on query-centric tasks, we compare it with selected eviction-based baselines: SNAPKV, $H_2O$, SIRLLM, INFINIPOT, and vanilla LOCRET. We also include FULLATTN (implemented with FLASH-ATTN) and MINFERENCE for reference. The RULER benchmark consists of 500 synthetic queries per task, each with a context length of 128K tokens. All methods are tested on `Llama-3.1-8B-instruct`.

For LOCRET-Q and LOCRET, we set the budget size $b$ to 6000, chunk size $B$ to 4096, stabilizers length $n_s$ to 2500, and local length $n_{loc}$ to 100. For SNAPKV, the voting window size is set to 100, with the last 100 tokens retained. For $H_2O$, due to its reliance on full-sequence attention scores, we use a layer-wise chunked prefill pattern with a chunk size of 1024. Larger chunk size would result in an out-of-memory error. For SIRLLM, we configure the start size to 4, recent size to 1000, and budget size to 6000. For INFINIPOT, we set the budget size to 6000 with 50% NuC ratio. All evaluations are conducted on a single NVIDIA A800-80GB GPU.

For prefill and decode speed testing, all methods except $H_2O$ are implemented with FLASH-ATTN; $H_2O$ uses PyTorch's vanilla attention due to its incompatibility with efficient attention implementations. The speeds are averaged on the first 5 entries of NIAH-Simple-1.

## B Evaluation on LongBench

We conduct additional experiments to evaluate Locret on LongBench (Bai et al., 2024b), comparing it with baselines such as Full Attention, MInference, InfLLM, and SirLLM. For this evaluation, we used `Phi-3-mini-128K` with a retained head trained on LongAlign. To ensure a fair comparison, we excluded all Chinese subtasks from LongBench and focused solely on the English subtasks, as `Phi-3-mini-128K` was not specifically trained on Chinese corpora. The results are presented below. For LOCRET , we follow the hyperparameters presented in Table 9.

We also report the maximum memory usage, including the GPU memory, the CPU memory, and the total maximum memory, alongside the average score on LongBench. For FULLATTN, we exclude the maximum memory usage, aligning with Figure 4.

From the experiments above, LOCRET demonstrates the best overall performance and excels in the majority of subtasks. It outperforms all the baselines without any noticeable performance degradation while consuming less memory. Although MInference also avoids performance drops, it requires more GPU memory compared to LOCRET. SirLLM achieves comparable memory usage but shows some performance decline compared to FULLATTN and LOCRET. InfLLM exhibits the most significant performance drop, and its offloading

Table 11: Comparison of methods on LongBench and memory usage.

| Method | LongBench | Max GPU Memory | Max CPU Memory | Total Max Memory |
|---|---|---|---|---|
| FULLATTN | 41.73 | - | - | - |
| MINFERENCE | 41.73 | 27.63 | 0.17 | 27.80 |
| SIRLLM | 40.51 | 18.29 | **0.05** | 18.34 |
| INFLLM | 32.95 | 20.03 | 8.95 | 28.98 |
| **Locret** | **42.31** | **17.71** | 0.15 | **17.86** |

mechanism results in the highest CPU memory usage, making it the method with the largest total memory consumption. These results highlight LOCRET as an outstanding approach for evaluation on LongBench.

## C   Training Robustness

### C.1   Intermediate Size of the retaining head

We align all the training settings as described in Section 5.1 and only change the intermediate size of retaining heads $d_{\mathbf{R}} \in \{256, 512, 1024, 2048, 4096\}$ with the backbone model `Phi-3-mini-128K`. The trained model is evaluated on L-Eval and we report the average L-Eval score corresponding to each intermediate size. Results are listed in Figure 6. The performance variations among all the settings are minimal compared to the changes in the intermediate size, surpassing all baselines in Table 2 and Table 3. This indicates that out method exhibits good performance stability regardless of the intermediate size of the retaining head $\mathbf{R}$s.

Table 12: L-Eval scores with different intermediate size of the retaining head $d_{\mathbf{R}}$. (Detailed)

| | `Phi-3-mini-128K` on L-Eval | | | | | | |
|---|---|---|---|---|---|---|---|
| $d_{\mathbf{R}}$ | CodeU | NQ | CUAD | NarrativeQA | QMSum | SPACE | Avg.↑ |
| 256 | 8.89 | 51.52 | 23.05 | 16.21 | 15.26 | 13.77 | 21.45 |
| 512 | 6.67 | 50.61 | 23.33 | 16.67 | 15.02 | 14.23 | 21.09 |
| 1024 | 8.89 | 51.49 | 22.23 | 16.42 | 14.86 | 14.06 | 21.33 |
| 2048 | 7.78 | 54.09 | 21.91 | 16.46 | 15.00 | 13.89 | 21.52 |
| 4096 | 10.00 | 52.33 | 23.52 | 16.15 | 14.81 | 14.02 | 21.81 |

We train different retaining head $\mathbf{R}$s with $d_{\mathbf{R}} \in \{256, 512, 1024, 2048, 4096\}$. We keep all the other hyperparameters same, and train on the same dataset. From Table 12, LOCRET shows stability to the intermediate size, in both overall performance and the performance of each single task. While increasing the intermediate size, we observe very slight overall performance enhancement. However, the performance variance is negligible compared to the increase of parameter size, thus we choose to maintain the intermediate size in a small scope to take balance of performance and efficiency.

### C.2   Training Data Insensitivity

We also consider the sensitivity of the training data, which leads us to ablate the training dataset by training on LongAlign (Bai et al., 2024a) and Anti-Haystack (Pan, 2024), comparing these results with those from LongAlpaca (Chen et al., 2024) in the original training setting. We also align other settings to the original setting and choose the backbone model to be `Phi-3-mini-128K`. We report the average L-Eval score for each training dataset. The results in Table 8 shows that LOCRET has high insensitivity towards different training data. The performance impact of different data recipes is minimal, indicating that our method can be trained on any long-context tuning dataset.

Table 13: L-Eval scores of LOCRET trained on various dataset. (Detailed)

| Phi-3-mini-128K on L-Eval | | | | | | | |
|---|---|---|---|---|---|---|---|
| Dataset | CodeU | NQ | CUAD | NarrativeQA | QMSum | SPACE | Avg.↑ |
| LongAlpaca | 8.89 | 51.49 | 22.23 | 16.42 | 14.86 | 14.06 | 21.33 |
| LongAlign | 10.00 | 55.13 | 21.34 | 16.40 | 15.01 | 14.09 | 22.00 |
| Anti-Haystack | 8.89 | 52.91 | 20.87 | 13.73 | 13.84 | 14.10 | 20.72 |

Table 14: L-Eval scores of LOCRET trained on different tasks.

| Phi-3-mini-128K on L-Eval | | | | | | | |
|---|---|---|---|---|---|---|---|
| Dataset | Code CodeU | QA NQ | CUAD | NarrativeQA | Summarization QMSum | SPACE | Avg.↑ |
| LCC (Code) | 7.78 | 56.87 | 14.01 | 20.93 | 15.10 | 13.87 | 21.43 |
| Trivia-QA (QA) | 8.89 | 56.77 | 14.15 | 20.49 | 14.75 | 14.03 | 21.51 |
| Gov-Report (Summ.) | 4.44 | 57.18 | 14.16 | 21.66 | 15.53 | 13.40 | 21.06 |

We conduct training on various datasets and benchmark the trained weights on L-Eval with `Phi-3-mini-128K` backbone, to show the stability towards training datasets. For each datasets, we set the training hyperparameters same and truncate the context to 10240 tokens. We train the first 3000 steps of LongAlpaca and LongAlign. Since Anti-Haystack is a relatively smaller dataset, we utilize the whole dataset, which consist of 2424 entries. According to (Chen et al., 2024; Bai et al., 2024a; Pan, 2024), all three datasets are general long-context SFT datasets, consisting of a mixed collection of diverse long-context tasks such as long-document question answering, summarization, and code generation. Therefore, the primary differences among these datasets lie in the relative proportions of various task types and the data quality of individual documents. The results in Table 13 shows that different training dataset recipe exhibits minor effect towards the overall performance. LOCRET can obtain competitive performance without delicately selecting the training data. We further perform a cross-task evaluation to demonstrate the generalizability of LOCRET, as shown in Table 14. Specifically, we consider three representative long-context tasks: coding, question answering (QA), and summarization. The model is trained on LCC[1] for coding, TriviaQA (Joshi et al., 2017) for QA, and GovReport (Huang et al., 2021) for summarization. As presented in Table 14, a model trained on one task can be effectively transferred to others. For QA and summarization, the transfer is nearly seamless, while the coding task shows a modest performance drop when trained without code data. However, models trained on coding data still generalize well to other tasks. Overall, LOCRET exhibits strong cross-task transferability and adaptability across diverse long-context domains.

## D  Ablation on the Smoothing Loss

We incorporate a smoothing loss, denoted as $\mathcal{L}_2$, into the training objective of LOCRET (Equation 6), which is applied to the predicted CIS between adjacent positions. This design encourages locality in predictions, a property commonly observed in natural language. To examine the effect of the smoothing loss, we evaluate models trained with and without $\mathcal{L}_2$ on the R.Number task of ∞Bench, where the target number appears in a locally concentrated region within a noisy document. As shown in Table 15, removing $\mathcal{L}_2$ leads to a significant performance degradation, confirming the effectiveness of the smoothing constraint. One illustrative failure case is when the model predicts "810007779" instead of the correct "8100077779", missing one "7". This example highlights that without enforcing locality, the model tends to lose information from crucial regions.

---

[1] https://huggingface.co/datasets/microsoft/LCC_python

Table 15: Ablation on the smoothing loss $\mathcal{L}_2$.

| Phi-3-mini-128K on R.Number | |
|---|---|
| Dataset | R.Number |
| LOCRET | 97.46 |
| LOCRET w/o $\mathcal{L}_2$ | 79.15 |

## E   Orthogonality to Other Methods

Table 16: Quantization with FULLATTN and LOCRET. "M" represents Method and "$-\Delta$" represents the gap of average L-Eval score.

| Setting | M | M-4bits | $-\Delta$ |
|---|---|---|---|
| M=FULLATTN | 29.08 | 28.52 | 0.56 |
| M=LOCRET | 27.96 | 27.11 | 0.85 |

Table 17: The average L-Eval scores of Lo-CoCo, LOCRET, and the combination of Lo-CoCo and LOCRET.

| Method | LoCoCo | LOCRET | **Combination** |
|---|---|---|---|
| L-Eval | 26.01 | 27.96 | **28.70** |

**KV cache quantization.** According to Zhang et al. (2024c), eviction-based methods like $H_2O$ struggle with compatibility when combined with KV cache quantization. Quantization introduces significant disturbance in the estimation of heavy-hitters, leading to severe performance degradation. However, LOCRET is not affected by such issues and can be combined with quantization while maintaining most of its performance. Here, we compare the performance degradation caused by quantization on LOCRET with that of the full attention method using the same metrics. We use Quanto as the quantization backend and report the average L-Eval score with `Llama-3.1-8B-instruct` as the model backbone. Table 16 shows that the performance drop caused by quantization on LOCRET is only slightly higher than that observed with the full attention method, indicating that LOCRET is a quantization-friendly approach. More details of the experiment are provided in Appendix E.1.

**Token merging.** As described in Section 2, token dropping can also be implemented through an attention pool. Attention pool-based methods (Xiao et al., 2024c; Cai et al., 2024a; Mu et al., 2024; Munkhdalai et al., 2024) merge adjacent tokens or cache units into an attention pool, maintaining a static cache size. These methods are orthogonal to LOCRET , as the evicted tokens can be merged into a small cache pool and retained in GPU memory. We conduct the following experiment to demonstrate that LOCRET can serve as an effective plug-in scoring function within such frameworks, enhancing performance without increasing memory budget. We select LoCoCo (Cai et al., 2024a) as a representative of the latest attention pool-based methods. LoCoCo maintains a cache set consisting of two parts: the heavy hitters and the convolved non-heavy hitters. During each chunked prefill step, LoCoCo first identifies a set of heavy hitters according to $H_2O$ (Zhang et al., 2024d), then applies 1-D convolution to the non-heavy hitters to compress them into a static size. By replacing $H_2O$'s heavy-hitter scoring function with LOCRET, we retain the cache units with high CIS and convolve the others. We compare this combination with standalone LoCoCo and LOCRET on L-Eval using the `Llama-3.1-8B-instruct` backbone and report the average score across all selected tasks. As shown in Table 17, LOCRET achieves a higher score than LoCoCo, and the combined algorithm outperforms both standalone methods. This suggests that LOCRET provides a more accurate scoring function compared to $H_2O$, and the two methods complement each other, demonstrating their orthogonality. Further details of the experiment are provided in Appendix E.2.

**Head-wise Budget Allocation.** Since LOCRET evict cache units across the attention heads independently, it is compatible with head-wise budget allocation. Here, we combine LOCRET with PYRAMIDKV (Cai et al., 2024b). PYRAMIDKV assumes that identifing the important cache in deeper layers are simpler than shallow layers, thus it allocates more budget to the shallow layers. We evaluate LOCRET+PYRAMIDKV on the following subtasks of $\infty$Bench using `Phi-3-mini-128K`. Results presented in Figure 18 shows the compatibility of the two methods.

Table 18: ∞Bench scores of the combination of LOCRET and PYRAMIDKV.

| | | | | | Phi-3-mini-128K on ∞Bench |
|---|---|---|---|---|---|
| Method | R.Number | E.Sum | E.MC | C.Debug | Avg.↑ |
| LOCRET | 97.46 | **16.82** | 46.29 | 29.71 | 47.57 |
| LOCRET+PYRAMIDKV | **99.66** | 15.82 | **48.03** | **30.00** | **48.38** |

## E.1 Combination with Quantization

Table 19: L-Eval scores of FULLATTN, FULLATTN-4bits, LOCRET and LOCRET-4bits. (Detailed)

| | | | | Llama-3.1-8B-instruct on L-Eval | | | |
|---|---|---|---|---|---|---|---|
| Method | CodeU | NQ | CUAD | NarrativeQA | QMSum | SPACE | Avg.↑ |
| FULLATTN | 10.0 | 66.84 | 38.91 | 23.11 | 18.76 | 16.86 | 29.08 |
| FULLATTN-4bits | 7.78 | 66.64 | 38.25 | 22.76 | 18.85 | 16.84 | 28.52 |
| LOCRET | 8.89 | 63.03 | 37.21 | 23.59 | 18.17 | 16.87 | 27.96 |
| LOCRET-4bits | 4.44 | 63.22 | 36.95 | 22.80 | 18.43 | 16.81 | 27.11 |

We compare the combination of LOCRET and HF-4BITS quantization with the full attention method and the standalong HF-4BITS quantization. We utilize the official implementation of Hugging Face, with Quanto as the backend of quantization. Other hyperparameters are kept same as described in Section 5.1. We conduct the experiment on L-Eval and report the average score, with `Llama-3.1-8B-instruct` backend. The results in Table 19 shows that the degradation caused by quantization is not significantly high, showing that LOCRET exhibits good robustness on data representation and it is friendly to quantization.

## E.2 Combination with LoCoCo

(a) L-Eval scores of LOCOCO, LOCRET and the combination LOCOCO+LOCRET. (Detailed)

| | | | | Llama-3.1-8B-instruct on L-Eval | | | |
|---|---|---|---|---|---|---|---|
| Method | CodeU | NQ | CUAD | NarrativeQA | QMSum | SPACE | Avg.↑ |
| FULLATTN | 10.0 | 66.84 | 38.91 | 23.11 | 18.76 | 16.86 | 29.08 |
| LOCOCO | 4.44 | 61.10 | 35.84 | 19.83 | 18.15 | 16.71 | 26.01 |
| LOCRET | 8.89 | 63.03 | 37.21 | 23.59 | 18.17 | 16.87 | 27.96 |
| LOCOCO+LOCRET | 7.78 | 66.33 | 38.01 | 24.85 | 18.31 | 16.92 | 28.70 |

We compare the combination of LOCOCO and LOCRET with the standalone methods. For LOCOCO, we train the convolution head with the size of convolved cache set to 2048. We extend the context length through chunked prefill training to 64K, which is longer than all tasks' average input length. The convolution kernel is set to 21, and we train the newly-added convolution and layer norms for 200 steps, following the original setting. Since the original `Llama-3.1-8B-instruct` supports 128K context length, we do not modify its positional embedding. During Inference, we keep a cache budget size of 16384. In the standalone LOCOCO setting, there are 2048 cache units are convolved, while the others are the heavy-hitters selected by $H_2O$. In the combined algorithm, we replace $H_2O$ to LOCRET. We select 14336 cache units with the highest CIS, and convolve the other evicted tokens into 2048 cache units. In all methods, we set the local length to 0, following the original setting.

## F Extremely Long Context Evaluation

We create a dataset similar to ∞Bench's R.Number, with an average length of 10 million tokens. Each data point contains a 10-digit number string inserted into an irrelevant context, and the task is to retrieve

the inserted number. The dataset consists of 50 examples, with the number strings uniformly distributed throughout the context. We used the hyperparameters from Table 9, with the exception of setting the chunk size to 10240 to speed up inference. The results, presented below in Table 21, show that Locret can efficiently process extremely long contexts. In this experiment, the cache budget is set to 6000, and the compression ratio is $1747.6\times$.

Table 21: Inference speed with Retaining Heads.

| `Phi-3-mini-128K` on 10M context | |
|---|---|
| Dataset | R.PassKey_10M |
| LOCRET | 100.00 |

## G Compressing Multi-turn Conversations

Compared to query-aware eviction methods, such as SNAPKV (Li et al., 2024b), LOCRET is a more suitable solution for multi-turn conversation scenarios. This is because the evaluation of cache importance in LOCRET is based on the cache itself, rather than being dependent on the subsequent query. To demonstrate this, we evaluate LOCRET on the Rock-Paper-Scissors benchmark introduced in SIRLLM (Yao et al., 2024). Since SIRLLM is specifically designed for such scenarios, we use it as our baseline in this benchmark. Results in Table 22 show that Locret is also effective in multi-turn conversation contexts.

The hyperparameters are aligned with those used in SIRLLM, with the cache budget set to 1024, and no stabilizers are retained, as SIRLLM does not retain local tokens in this benchmark. We perform 2000 turns as same as the original SIRLLM settings. The results are presented below.

Table 22: Rock-Paper-Scissors scores of LOCRET and SIRLLM.

| | `Phi-3-mini-128K` on Rock-Paper-Scissors | | | | | | | | | | |
|---|---|---|---|---|---|---|---|---|---|---|---|
| Preference | Rock | | | Paper | | | Scissors | | | Avg. | |
| | win | tie | lose | win | tie | lose | win | tie | lose | win↑ | lose↓ |
| SIRLLM | 40.00 | 31.75 | 28.25 | 27.5 | 36.55 | 35.96 | 29.35 | 25.15 | 45.50 | 32.28 | 36.57 |
| **Locret** | 18.95 | 50.00 | 31.05 | 30.35 | 19.45 | 50.20 | 52.05 | 27.25 | 20.70 | **33.78** | **33.98** |

## H Discontinuous Context and Stablizers

Evicting cache units results in context discontinuity, which causes unstable CIS prediction and inaccurate calculation of later tokens. Thus, we always retain the stabilizers, which are consist of the last $n_s$ cache units in each chunked prefill step. We ablate $n_s$ on R.Number of $\infty$-Bench in the proposed algorithm to demonstrate the necessity of incorporating stabilizers in the design. The results in Figure 3a show that lower stabilizer length $n_s$ causes severe performance degredation and the model fails completely when the stabilizers are absent. We report the maximum absolute error of the last hidden state of the input prompt across different layers in Figure 3b. Large errors can be observed when the stabilizers are short or absent. We also report the mean absolute error of the predicted causal importance values with different stabilizer lengths, compared to the case without evicting any cache units, in Figure 3c. We also observe high errors when the stabilizer length is limited. This explains the reason for failure when the stabilizers are short or absent: context discontinuity leads to instability in the prediction of CIS, resulting in errors during cache eviction and amplifying errors in the hidden states.

## I Retaining Heads Do not Slow Down Inference

We evaluate the model's forward throughput under varying context lengths, both with and without retaining heads. The results are summarized below in Table 23. "**R**" represents the retaining heads, and the throughput is reported in tokens per second (tok/s) in the format "Avg. / Std."

Table 23: Inference speed with Retaining Heads.

| Context Length | 1024 | 2048 | 3072 | 4096 |
|---|---|---|---|---|
| w/o $\mathbf{R}$ Speed | 18674 / 443 | 19743 / 464 | 19982 / 402 | 20304 / 187 |
| w/ $\mathbf{R}$ Speed | 17118 / 1117 | 18503 / 546 | 19054 / 283 | 19153 / 174 |

From the results, no significant latency increase is observed when using retaining heads. The numerical differences are attributed to systematic variations rather than additional overhead introduced by retaining heads during inference.

## J Retained Patterns of Locret

We investigate the retained patterns of LOCRET. We trace the cache units at each attention head through the chunked prefill on R.Number, M.find and E.MC of $\infty$Bench with backbone `Phi-3-mini-128K`, and investigate the pattern variation among different layers on R.Number. We display the results in Figure 7 and Figure 8. The yellow parts are the retained cache, where the y-axis represents cache position and x-axis is the time axis.

Figure 7 shows that the pattern is mostly decided by the tasks, where both heads shows similar pattern in the same task. In R.Number, we are able to observe a strong signal between token 10000 and 15000, which is the position of the inserted number string, indicating that LOCRET can identify the potentially answer-related parts by giving high predicted values of CIS. In M.Find, we can observe the StreamingLLM (Xiao et al., 2024c) pattern, where the tokens at the beginning of the sequence are always important. This is also mentioned as the Λ-pattern in MINFERENCE. We can also discover the vertical lines in the middle of the sequence. This pattern is also approached by MINFERENCE (Jiang et al., 2024a) by the pattern "vertical-and-slash". In E.MC, $H_2O$ (Zhang et al., 2024d) and ScissorHands (Liu et al., 2024a) pattern can be observed, following the assumption that if a token is activated at some point, it will continue to be activated in the consequencing process. Noticing that the vertical lines always come in groups, which is the fundament of INFLLM (Xiao et al., 2024a) retrieving blocks to calculate. The comparison between two heads also shows that different heads exhibits different features. Head 22 of layer 11 shows stronger vertical lines at some point, where retained pattern of head 14 layer 11 is more even. Head 14 of layer 11 also gives stronger signal to the initial tokens, where this effect is less strong in head 22 layer 11. We also conduct experiments to investigate the patterns across layers. In Figure 8, we show that the pattern variance of the same head in different layers can be large. In shallow layers, e.g. layer 1 and 5, the retained cache units appears to be periodical and semantic independent. However, in middle layers, e.g. layer 13 and 17, the position of the inserted number string is strongly highlighted, indicating that semantic takes over to be the dominant factor. In the deepest layers, e.g. 21, 25 and 29, the highlighted vertical line at the position of the inserted string becomes more accurate.

The retained pattern at different layers shows various features, which might be a good handle to investigate how LLMs understand and process natural language queries.

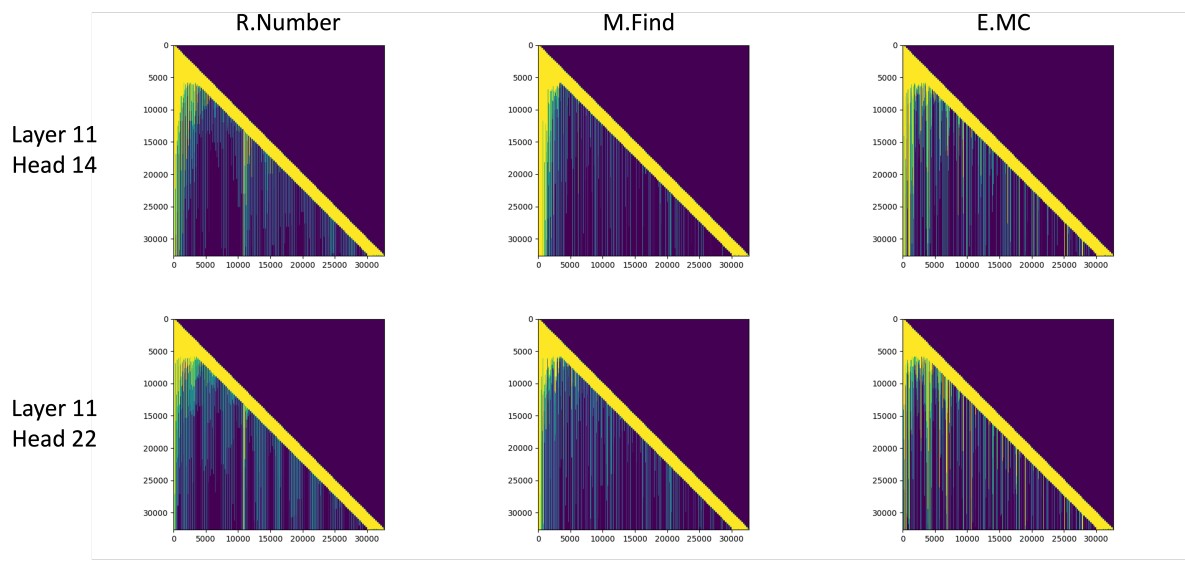

Figure 7: Head patterns across multiple tasks.

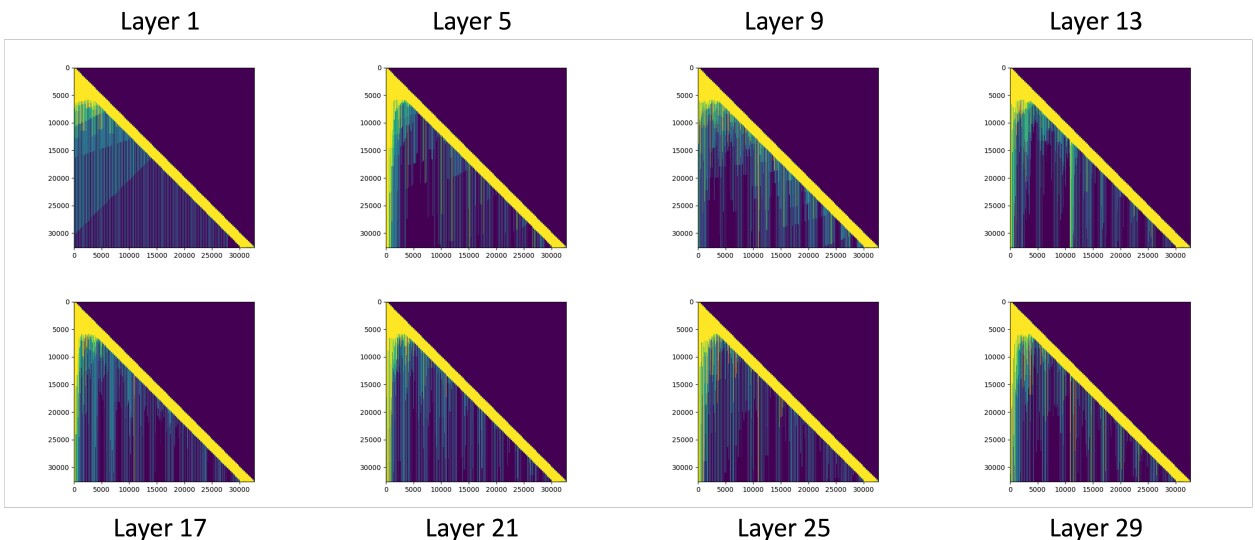

Figure 8: Layer patterns of R.Number

