# OpenReview forum: "Locret: Enhancing Eviction in Long-Context LLM Inference with Trained Retaining Heads on Consumer-Grade Devices"
_TMLR — Accepted by TMLR_

### Review · Reviewer_otMS · 2025-10-09

**Summary Of Contributions:**

### Summary

The paper addresses the problem of efficiently evicting KV caches for long-context inference in large language models in causal settings, such as streaming or chunked prefill. The authors argue that most existing eviction strategies (e.g., H2O, SnapKV) implicitly rely on global importance estimates that are not available in causal settings. To quantify this mismatch, they present a consistency metric that compares global (full-context) and local (causal) token importance scores, demonstrating that existing strategies are unstable across the two regimes. To address this, they propose Locret, a causally trained importance scorer (CIS) that predicts future token relevance using only past attention, implemented via Retaining Heads. The method aims to minimize the discrepancy between local cache scores and ideal full-context scores.

In the experiment, the paper trained Locret on the LongAlpaca dataset. It evaluated the proposed method across multiple benchmarks. The results show that Locret not only matches or exceeds full-attention baselines in accuracy but also provides significant memory savings and speed up.


### Strengths

1. Clear and well-motivated diagnosis of why prior eviction methods fail in streaming/chunked inference.
2. The method requires minimal training and is relatively lightweight.
3. Strong and extensive empirical results. The paper experiments on two models and two benchmarks, comparing with many baselines.


### Weaknesses

1. Generalization of the trained Retaining Heads across tasks is implied but not extensively validated.
    1. The causal training objective relies on future attention during supervision. It would be clearer to discuss whether this introduces bias towards the datasets used to train the Retaining Heads.
    2. Table 12 shows that there are some inconsistencies between training datasets. Extending this experiment and using statistics to quantify the "insensitiveness" would strengthen the paper.
2. Although it is a good sign, some results outperforming full attention lack discussion. These results introduce a contradiction between the consistency (Figure 1) and the benchmark performances (Tables 2 and 3).
3. The CIS training objective includes a smoothing / L2 regularization term, but no ablation is provided to isolate its contribution. Is the performance gain primarily due to the Retaining Head architecture, or does the smoothness constraint dominate? A comparison of CIS trained with and without the regularization term would clarify this.
4. In general, the paper is well organized and provides a brief introduction to the extra materials in the appendix. However, Section 3 would be clearer if the paper introduced preliminary (4.1) and included the limitations of the causal inference cache.

**Audience:**

Yes

**Audience Explanation:**

This paper addresses a crucial problem in making LLMs more efficient.

**Broader Impact Concerns:**

No broader impact concerns.

**Claims And Evidence:**

No

**Claims Explanation:**

There are several claims made by the paper. The majority of them have experimental results to support their claims in the chunked prefilled setting.

1. Locret uses less memory while maintaining performance (Section 5.2)
2. Locret achieves a higher speed-up on a small GPU (Section 5.3)
3. Locret can handle query-driven task (Section 5.4)
4. Locret requires stabilizer (Figure 3)

However, some implicit claims would need more evidence:

1. Locret robustness (Section 5.5)
    > From our observations, the training of Locret exhibits strong robustness. Despite changes in $d_R$ and the dataset, the performance variations shown in Figure 6 and Table 7 are minimal.
    The robustness of different training data is not clearly shown in Table 7 (as there are only three data points). Upon closer examination of Table 12, several inconsistencies are evident from task to task across different training datasets.
2. Locret requires $L_2$. While the stabilizer is shown to be important, it is unclear whether the $L_2$ is needed.

**Requested Changes:**

1. Clarify the training and generalization behavior of Retaining Heads. The authors can include one of the following changes:
    - Provide a cross-task generalization experiment (e.g., train on dataset A, evaluate on dataset B without retraining) to support the claim that Locret is robust to training distribution shifts.
    - Expand Table 12 into a broader study with mean/variance statistics to better quantify robustness rather than relying on a few manually selected data points.
2. Some benchmark results exceed full attention, which appears contradictory given that consistency metrics suggest Locret approximates rather than improves upon full attention. Please provide a short discussion explaining this effect.
3. Include an ablation on the $L_2$ smoothing.
4. Improve clarity in Section 3. The definition of the consistency metric is clear. Still, the narrative would be more straightforward to follow if Section 4.1 (preliminary formulation of local vs. global scoring) were introduced earlier or briefly referenced. Additionally, explicitly stating the limitations of causal scoring before presenting consistency results would help motivate why inconsistency matters.

---

> ### Author Response · Authors · 2025-10-21
> **Response to Reviewer otMS’s Comments**
>
> We sincerely thank you for providing such detailed reviews and suggestions. According to the feedback, we have updated our manuscript following the Requested Changes. The modified parts are marked in blue.
>
> 1. To clarify the training and generalization behavior of the Retaining Heads, we conduct a cross-task experiment on Page 20. We train and evaluate Locret on coding, question-answering, and summarization tasks. The results reported in Figure 13 show that Locret demonstrates good generalization ability. We also add a short discussion on the differences among the three datasets used in Figure 12 for better clarity.
>
> 2. The reason why some benchmark results of Locret surpass those of FullAttn is due to its noise-removal effect: irrelevant tokens are evicted, allowing the model to focus more on answer-related parts. We add a short discussion of this on Page 9.
>
> 3. An ablation study on the smoothing loss is now presented in Appendix D (Page 21). The design of the smoothing loss aims to enhance prediction locality, and we observe a large performance degradation on R.Number (a hard retrieval task) when the smoothing loss is removed.
>
> 4. To clarify the original Section 3, we have moved the preliminary section before it and formed a new Background Section (Section 2). In this section, we describe the inconsistency in existing methods and state our motivation for introducing a consistent and causal importance score. Such inconsistency in previous methods is undesirable, as it prevents their application in chunked prefill scenarios.

---

> ### Author Response · Authors · 2025-10-27
>
> We sincerely appreciate your thoughtful and detailed reviews once again. If there are any remaining questions or concerns, we would be very grateful for the opportunity to address them. We believe that the current revision has made sincere efforts to respond to all the requested changes, and the new experiments aim to further support our claims. If anything remains unclear, we would be glad to provide additional explanations or supplementary experiments.

---

### Review · Reviewer_Fwwc · 2025-10-21

**Summary Of Contributions:**

This paper introduces Locret, a lightweight, training-based framework for key-value (KV) cache eviction in long-context LLM inference. It tackles the fundamental challenge of scaling LLMs on consumer-grade devices, where memory constraints limit long-context processing.
The core innovation lies in the use of trainable “retaining heads” that learn a causal importance score (CIS) for each cache unit, enabling precise and causal cache eviction compatible with chunked prefill. Locret requires minimal additional training (<1 GPU hour) and can be seamlessly integrated into existing decoder-only architectures.
A variant, Locret-Q, extends the approach to query-driven tasks by introducing query-aware eviction. Extensive experiments on ∞Bench, L-Eval, LongBench, and RULER show that Locret achieves:

- Up to 20× compression of the KV cache with <10% quality loss.

- 128K+ context inference on a single NVIDIA 4090 GPU.

- State-of-the-art tradeoffs between accuracy, memory, and inference speed, outperforming strong baselines like H2O, SnapKV, SirLLM, and MInference.

Overall, Locret offers a practical, principled, and efficient approach to democratizing long-context LLM inference.

**Audience:**

Yes

**Audience Explanation:**

- The method is algorithmically novel (introducing trainable retaining heads and causal importance scoring).

- In addition, it is systemically practical (lightweight integration, low compute overhead).

Overall, the proposed framework bridges the gap between model design, memory optimization, and system deployment, making it of strong interest to readers working on efficient inference, compression, and scalable reasoning in LLMs.

**Broader Impact Concerns:**

There are no major ethical red flags in this work.

**Claims And Evidence:**

Yes

**Claims Explanation:**

The paper provides rigorous empirical validation and clear theoretical motivation:

- It systematically analyzes the local-global discrepancy in existing scoring functions (e.g., H2O, SnapKV) and demonstrates Locret’s superiority in maintaining consistency without access to future tokens.

- The architectural design (retaining heads, causal CIS training) is well-motivated, with ablations isolating each component (budget size, stabilizer length, chunk size, training dataset, intermediate size).

- Benchmark results across multiple long-context datasets consistently show Locret’s performance advantages in both efficiency (memory/speed) and accuracy, especially under realistic consumer-grade settings.

- Experimental transparency is high: training time, hardware settings, and hyperparameters are all provided, and comparisons include a wide range of state-of-the-art baselines. The empirical evidence thus fully supports the paper’s claims, and the methodology is both replicable and technically sound.

**Requested Changes:**

None

---

> ### Author Response · Authors · 2025-10-21
> **Response to Reviewer Fwwc’s Comments**
>
> We sincerely thank you for your support and for providing such detailed reviews. We will continue our efforts in this direction to further advance efficient long-context inference.

---

### Review · Reviewer_4ga9 · 2025-10-26

**Summary Of Contributions:**

The paper proposes a novel eviction-based strategy for key-value (KV) cache optimisation with the goal of making long-context LLM inference more efficient. This strategy is called Locret and is based on training a separate set of retaining heads that predict an importance score of elements in the KV cache. The retaining heads are trained on a smaller, shorter-context datasets by supervising the desired importance score with the maximum attention score over the answer produced from a KV cache prefilled on the prompt. The heads are then used at generation time to prefill the KV cache up to its memory limit by removing those pairs with a low importance score. Consequently, combined GPU and CPU memory never exceeds the desired memory budget once the budget is set. Locret is empirically verified on a number of long-context benchmarks against other memory reduction techniques.
#### Strengths
1. **The explanation of the method is clear.** The entire pipeline that Locret is building is clearly explained. In particular, the exposition is well-written throughout. The problem that is being solved is clearly demarcated as is its solution. I also appreciate a sufficient amount of preliminaries for the reader to catch up on the specifics of the topic as well as related work.

2. **The method is intuitive and easy to understand, and the authors highlight they had to solve some issues.** The idea of training a separate module that learns to predict a meaningful score for retaining key-value pairs is certainly interesting and makes intuitive sense. However, it is certainly not intuitive that this strategy of building a KV cache *without* LLM retraining still works. From their exposition, the authors clearly had to optimise this idea for practical success and had to deal with a number of engineering challenges, e.g. adding stabilisers.
#### Weaknesses
1. **The writing could be augmented with more explanation of why choices were made.** Locret makes some choices in its pipeline, e.g. an MLP as retaining head or training on QA datasets with maximum attention score as target, without explaining why. This is a minor weakness and aimed to further improve the exposition.
2. **The experimental section does not provide convincing evidence for Locret.** See my specific concerns on the matter below.

I want to highlight here that Locret does *seem* to show promise with reasonable and interesting results, but that this can not be made certain yet. The method is interesting and only requires a more consistent and complete experimental section to show that it is impactful.

**Audience:**

Yes

**Audience Explanation:**

I think this work can be interesting for those members of the community that aim to optimise the efficiency of LLM inference as well as their practical ramifications.

**Broader Impact Concerns:**

I do not have any broader ethical concerns.

**Claims And Evidence:**

No

**Claims Explanation:**

While the method is explained well, it still hinges on a very convincing experimental section in order to be of any impact. However, I have a number of concerns with the experimental section.
1. **No variability metrics are reported.** All reported metrics are a single, average value without any notion of how variable those averages are. Please use variability metrics in order for the reader to gain an understanding of how statistically significant these results are. *Without variability metrics, reviewers could be looking at statistical noise instead of significance.*

2. **No consistent choice of baselines across experiments and missing baselines/ablations.** I am failing to see why there is not consistent choice of baselines across all experiments. For example, the evaluation on $\infty$Bench and L-Eval compares to InfLLM, 2-bit quantisation, SirLLM and MInference, while the query-driven experiment does not consider InfLLM, SirLLM or quantisation and adds SnapKV, H$_2$O and InfiniPot. Why are not all these methods used across the board? Additionally, it seems two important baselines are missing; higher-bit quantisation (at least 4-bit) and an ablation baseline that only utilises stabilisers. First, a comparison to 4-bit quantisation is important because it is much more commonly used than 2-bit quantisation. Even more, to avoid the overhead from inverse quantisation, why not quantise the LLM as well as, again, is more commonly done in practice? Second, a comparison to a baseline with a KV cache that *only* uses stabilisers (so only the last tokens fitting in memory) is important because of the importance stabilisers show to have for Locret too. In particular, Figure 5b shows that when the stabilisers fill the entire budget (6000) and there is no selection made by the retaining heads, the performance suddenly increase substantially. *Without these two baselines, the impact of Locret is not strongly supported.*

3. **Overly strong or wrong experimental conclusion in some places.** First example at the end of Section 5.2, paragraph 2: "In the shorter context scenario (L-Eval), a similar phenomenon is observed." A similar phenomenon is most certainly not observed from Figure 4 as all methods, except perhaps HF-2bits on Llama-3.1 actually *increase* both the GPU and total memory cost. This observation is very counter-intuitive. *Could you please elaborate?*
   Second example in the second paragraph of Section 5.3: "the quantization and dequantization processes become the bottleneck, leading to significantly slower speed performance". It does not lead to slower speeds. Table 4 shos that quantisation with HF-2bits is even quicker for Llama-3.1, it is the performance that tanks for quantisation. *Can you elaborate why this is the case for Llama-3.1 but not for Phi3?*

4. **Reporting of strange or incomparable metrics as well as weird testing practices.** Example 1 is Table 4; there is no need for an arbitrary "success" checkmark. The accuracy speaks for itself. Example 2 is the "solution" to OOM of some methods in Section 5.3: "we remove some tokens from the middle of the input sequence in those cases, marking these settings with ∗,". Since the task here is to recover a passkey from a large amount of irrelevant text, the removing of tokens might remove the passkey to find. Additionally, no precise explanation is given *how* it is decided which tokens get removed. Example 3, perhaps the most incorrect practice, is the *limiting of testing to the first 20 entries of the test set if bad performance is observed*. This way of testing only produces numbers that are *incomparable* as they are computed from different test sets. Poor performance is no reason to not report comparable metrics throughout (Tables 5 and 6).

**Requested Changes:**

In accordance with the above raised issues, I request the authors to
1. Add variability metrics. Preferably those with less distributional assumptions, e.g. quartiles/quantiles.
2. Make set of baselines consistent across all experimental sections. Add 4-bit quantisation (with quantised LLM) as well as baseline with only stabilisers.
3. Check the experimental section for errors and rewrite to be more faithful.
4. Solve or explain the use of strange testing practices.
5. Locret-Q should be part of the method (Section 4) and should not suddenly appear in the experimental section.

---

> ### Author Response · Authors · 2025-11-04
> **Response to Reviewer 4ga9’s Comments (Part 1)**
>
> We sincerely appreciate the time and effort you devoted to reviewing our paper and providing such detailed feedback. We have revised the manuscript according to your suggestions, and all corresponding changes are highlighted in red. Below, we provide a brief response to your comments.
>
> ---
> ### **Weakness**
>
> 1. **The writing could be augmented with more explanation of why choices were made.**
>
> > an MLP as retaining head
>
> We adopt a lightweight MLP to predict the importance of each KV pair because MLPs are expressive function approximators and introduce minimal inference overhead. Since the prediction only depends on a hidden state, this task is simple enough for a small MLP to learn effectively and achieve low training loss. We add a brief explanation on Page 5, section 4.2.
>
> > training on QA datasets with maximum attention score as target
>
> During training, we use max pooling because, in long-context QA tasks, not all answer tokens are strongly correlated with essential content (e.g., tokens like “a” or “the”). Max pooling helps filter out these irrelevant tokens and focus on the most informative signals.  We add a brief explanation on Page 6, section 4.2.
>
> 2. **The experimental section does not provide convincing evidence for Locret.**
>
> See below.
>
> ---
>
> ### **Requested Changes and "Explain your answer above" Section**
>
> 1. **Add variability metrics.**
>
> We use greedy decoding (temperature = 0) for all benchmarks. As a result, methods that do not involve training produce deterministic outputs with no randomness. Since only our method requires lightweight training, all other baselines should have zero variance. To verify this, we run each method three times and report the variance in Table 2 and Table 3. For Locret, we train the model for 3 times, and evaluate the trained model in greedy decoding mode.
> For example, when running Phi-3-mini-128K on the NQ task from L-Eval, the three runs produce the following results:
>
> | Methods | FullAttn | InfLLM | HF-2bits | SirLLM | MInference | Locret |
> | :-: | :-: | :-: | :-: | :-: | :-: | :-: |
> | Run-1 | 59.14 | 34.32 | 1.69 | 37.92 | 25.21 | 53.38 |
> | Run-2 | 59.14 | 34.32 | 1.69 | 37.92 | 25.21 | 52.24 |
> | Run-3 | 59.14 | 34.32 | 1.69 | 37.92 | 25.21 | 51.49 |
> | Mean | 59.14 | 34.32 | 1.69 | 37.92 | 25.21 | 52.37 |
> | Variance | 0.00 | 0.00 | 0.00 | 0.00 | 0.00 | 0.95 |
>
> We have included all variance statistics in Table 4 and Table 5. Please refer to the updated manuscript for details. The reported variances for Locret also confirm that its strong performance is not due to statistical noise.
>
> Notably, we are aligning our benchmark approach to the related works. In the papers of related works (SnapKV, InfLLM, MInference) and the papers of long-context benchmarks (InfiniteBench, L-Eval, RULER), all experiments are conducted in greedy decoding mode and no variance is reported. We believe that such benchmark approach is persuasive to show the capability of Locret.
>
> If more empirical results are needed, please raise your concerns and we are glad to conduct more experiments.

---

> > ### Author Response · Authors · 2025-11-04
> > **Response to Reviewer 4ga9’s Comments (Part 2)**
> >
> > 2. **Make set of baselines consistent across all experimental sections.**
> >
> > > I am failing to see why there is not consistent choice of baselines across all experiments. For example, the evaluation on Bench and L-Eval compares to InfLLM, 2-bit quantisation, SirLLM and MInference, while the query-driven experiment does not consider InfLLM, SirLLM or quantisation and adds SnapKV, HO and InfiniPot. Why are not all these methods used across the board?
> >
> > The reason why we select two sets of the baselines for Locret and Locret-Q is that they are designed for different scenarios.
> >
> > **Locret: General long-context tasks (non–query-driven).** For standard long-context inference, we compare against *a broader range* of KV-cache and attention optimization methods, including sparse attention, KV quantization, offloading, and eviction. Our goal is to show that Locret achieves higher performance while using equal or less GPU memory. In this setting, we focus on performance–memory trade-offs. Also, methods such as SnapKV and H2O are not included as primary baselines because their original papers show they are less effective than MInference under these conditions in MInference original paper.
> >
> > **Locret-Q: Query-aware tasks (RULER-like).**
> > Locret-Q specifically targets the challenge where *eviction-based KV cache methods are commonly believed to be unsuitable for query-driven tasks* like those in RULER. As shown in Table 5 and Table 6, strong baselines such as SnapKV and InfiniPot experience severe performance drops that cannot be fully recovered even with query-aware eviction (placing the query at the front).
> > In contrast, Locret can handle this setting effectively with only a slight modification (placing the query first). Therefore, in this scenario, we compare mainly against eviction-based methods (SnapKV, H2O, SirLLM, InfiniPot), as the focus is on evaluating whether Locret overcomes the core limitation of *this class* of methods.
> >
> > > First, a comparison to 4-bit quantisation is important because it is much more commonly used than 2-bit quantisation. Even more, to avoid the overhead from inverse quantisation, why not quantise the LLM as well as, again, is more commonly done in practice?
> >
> > We have already included a comparison between Locret and 4-bit quantization in Appendix E (Table 19, Page 22). The results show that 4-bit quantization achieves higher accuracy than Locret, which is expected due to its lower compression ratio. However, we also demonstrate that Locret is fully compatible with 4-bit quantization, and the combined method outperforms other baselines. Therefore, we do not treat 4-bit quantization as a primary baseline for two reasons: (1) its compression ratio is significantly lower than that of Locret, and (2) it can be integrated with Locret rather than being an alternative.
> >
> > Regarding quantizing the LLM weights, we believe this is slightly beyond the scope of our work.
> > (1) In extra long-context scenarios, the KV cache dominates memory usage, while the model weights contribute far less. Therefore, our method primarily targets KV cache optimization.
> > (2) Weight quantization is orthogonal to KV cache optimization and can be combined with most KV cache optimization if desired. For this reason, we consider such experiments to be outside the main focus of this paper.
> >
> > > Second, a comparison to a baseline with a KV cache that only uses stabilisers (so only the last tokens fitting in memory) is important because of the importance stabilisers show to have for Locret too. In particular, Figure 5b shows that when the stabilisers fill the entire budget (6000) and there is no selection made by the retaining heads, the performance suddenly increase substantially.
> >
> > Figure 5b actually conveys the opposite meaning. When the number of stabilizers reaches the entire budget (6000), the performance suddenly ***drops*** significantly rather than improves. A lower QMSum score indicates worse task performance. We argue that Figure 5b clearly demonstrates what happens when only stabilizers are used: a substantial performance degradation occurs. This is because the model is unable to retain long-distance information in the KV cache: only the most recent tokens are preserved. This setup is very similar to StreamingLLM, a baseline that also exhibits much lower performance, as reported in both the InfLLM and MInference papers.

---

> > > ### Author Response · Authors · 2025-11-04
> > > **Response to Reviewer 4ga9’s Comments (Part 3)**
> > >
> > > 3. **Check the experimental section for errors and rewrite to be more faithful.**
> > >
> > > > First example at the end of Section 5.2, paragraph 2: "In the shorter context scenario (L-Eval), a similar phenomenon is observed." A similar phenomenon is most certainly not observed from Figure 4 as all methods, except perhaps HF-2bits on Llama-3.1 actually increase both the GPU and total memory cost. This observation is very counter-intuitive.
> > >
> > > We have rewritten the discussion in Section 5.2, paragraph 2, and provided a more detailed analysis. The reason why HF-2bits on Llama-3.1 exhibits increased memory usage is that, due to the use of Grouped Query Attention (GQA), the main bottleneck for shorter contexts (e.g., in L-Eval) shifts to runtime memory rather than KV cache size. In quantization-based methods, the *full* KV cache must be dequantized back to BF16 at every layer before performing attention. In contrast, Locret stores only a *partial* KV cache directly in BF16 on the GPU. As a result, the overall memory usage of 2-bit quantization can be slightly higher than Locret.
> > >
> > > > Second example in the second paragraph of Section 5.3: "the quantization and dequantization processes become the bottleneck, leading to significantly slower speed performance". It does not lead to slower speeds. Table 4 shos that quantisation with HF-2bits is even quicker for Llama-3.1, it is the performance that tanks for quantisation. Can you elaborate why this is the case for Llama-3.1 but not for Phi3?
> > >
> > > On Llama-3.1, HF-2bits achieves a decoding speed of 1365.51 tok/s, which is the ***slowest*** among all methods (InfLLM: 2287.66, SirLLM: 1589.75, Locret: 3209.10).
> > > HF-2bits* appears faster only because its inputs are truncated, resulting in shorter actual context lengths. In fact, the speed of HF-2bits is highly dependent on input length, as it introduces no sparsity in attention computation or memory access. We provide further explanation in Section 5.3, paragraph 2 (Page 11).
> > >
> > > 4. **Solve or explain the use of strange testing practices.**
> > >
> > > > Example 1 is Table 4; there is no need for an arbitrary "success" checkmark. The accuracy speaks for itself.
> > >
> > > We remove the "success" checkmarks.
> > >
> > > > Example 2 is the "solution" to OOM of some methods in Section 5.3: "we remove some tokens from the middle of the input sequence in those cases, marking these settings with ∗,". Since the task here is to recover a passkey from a large amount of irrelevant text, the removing of tokens might remove the passkey to find. Additionally, no precise explanation is given how it is decided which tokens get removed.
> > >
> > > We follow InfiniteBench’s original truncation strategy, which removes tokens from the middle of the input until the desired sequence length is reached. This approach is widely adopted in InfiniteBench, as many inputs exceed 200K tokens and are truncated to 128K before being fed into the model. In our experiments, we only modify the “maximum supported length” in InfiniteBench.
> > >
> > > The reason for using this setting is to construct a vanilla baseline, reflecting what would happen if methods are directly deployed under memory-constrained scenarios. In such cases, truncation is unavoidable to prevent out-of-memory (OOM) errors. For example, if additional mechanisms such as a sliding window are incorporated into MInference to reduce memory usage, then the modified approach can no longer be regarded as the original “MInference,” since its core mechanism is altered. Therefore, we provide this vanilla baseline for fair comparison. We have added an explanation in Section 5.3, paragraph 1, Page 10.
> > >
> > > > Example 3, perhaps the most incorrect practice, is the limiting of testing to the first 20 entries of the test set if bad performance is observed. This way of testing only produces numbers that are incomparable as they are computed from different test sets. Poor performance is no reason to not report comparable metrics throughout (Tables 5 and 6).
> > >
> > > RULER is a synthetic benchmark where the number of samples per task is configurable. Increasing the number of entries only improves the reliability of results (i.e., reduces randomness), without altering the task itself. To ensure comparability, we first generate 20 entries per task for all methods reported in Table 5 and Table 6. Under this setting, methods such as H2O, SirLLM, InfiniPot, and Locret show clear failure cases, as they achieve zero or near-zero scores on many tasks.
> > >
> > > Next, for a higher-precision evaluation, we increase the number of entries to 500 per task and compare only the stronger methods: FullAttn, MInference, SnapKV, and Locret-Q. The conclusion remains consistent. Locret-Q outperforms all eviction-based baselines in this setting. Since we now use a consistent number of entries within each table, the comparison is fair.

---

> > > > ### Author Response · Authors · 2025-11-04
> > > > **Response to Reviewer 4ga9’s Comments (Part 5)**
> > > >
> > > > 5. **Locret-Q should be part of the method (Section 4) and should not suddenly appear in the experimental section.**
> > > >
> > > > We add Section 4.4: Processing Query-Driven Tasks in the method section to introduce Locret-Q earlier.

---

> > > > ### Comment · Reviewer_4ga9 · 2025-11-23
> > > > **Appreciate the clarifications**
> > > >
> > > > I thank the authors for their extensive clarifications and briefly comment on their responses.
> > > >
> > > > 1. **Variability metrics and other experimental practices.** Thank you for sanity checking the variability of Locret. I see that the nature of the benchmark and the common practice of greedy decoding indeed removes variability. It confirms Locret's (greedy-decoding) performance is not due to statistical noise. My concerns about other practices, i.e. the truncation strategy and reporting in Tables 5 and 6, have also been clarified. Thank you for now separating the higher-precision results in Table 7 and also highlighting the potential problems of the often-used truncation strategy.  An additional comment on the disadvantages of the ubiqitous use of greedy encodings in benchmarks could also be informative, especially as many deployed models do not use fully greedy decoding. Such comments on the shortcomings of common testing practices are highly valuable.
> > > >
> > > > 2. **KV cache memory dominance and quantisation.** While I do agree that the KV cache will dominate memory, I am left a bit confused by the current quantisation baseline. If I understand correctly, you quantise the KV cache, but need to perform inverse quantisation because the model weights are not quantised, correct? This dequantisation introduced computational overhead that slows down decoding, as mentioned in the related work on page 2. I also imagine dequantisation introduces additional memory as well, so would quantising the weights also not lead to potential memory and computation reduction? I do agree that quantisation, in any form, is indeed orthogonal to Locret and is hence a less crucial baseline, but I would like to have my confusion clarified. Especially since it seems common to quantise and dequantise, while I would expect a fully quantised cache and model to remove the need for dequantisation and hence improve the memory footprint.
> > > >
> > > > 3. **I have misinterpreted certain information**. First, I did not realise that the QMSum was 'lower is worse', which removes my concerns for an all-stabiliser comparison. Second, I did mean that HF-2bits* was faster and less performant, and I mixed this up with the discussion on HF-2bits (non-star). Thank you for adding a clarifying sentence there.

---

> > > > > ### Author Response · Authors · 2025-11-23
> > > > > **Response to Reviewer 4ga9's "Appreciate the clarifications"**
> > > > >
> > > > > Thank you for your reply. Here are our response to the remaining questions.
> > > > >
> > > > > ---
> > > > >
> > > > > 1 **Variability metrics and other experimental practices.**
> > > > >
> > > > > >  An additional comment on the disadvantages of the ubiqitous use of greedy encodings in benchmarks could also be informative, especially as many deployed models do not use fully greedy decoding. Such comments on the shortcomings of common testing practices are highly valuable.
> > > > >
> > > > > We acknowledge this limitation in our experiment, and we have added a brief discussion on Page 9 (red lines).
> > > > >
> > > > > 2. **KV cache memory dominance and quantisation.**
> > > > >
> > > > > > If I understand correctly, you quantise the KV cache, but need to perform inverse quantisation because the model weights are not quantised, correct?
> > > > >
> > > > > Yes. The model weights and hidden states are not quantized. The $\mathbf{Q}_t$ metric at decoding step $t$ is not quantized. Since the attention mechanism operates between $\mathbf{Q}_t$ and the existing KV cache, the KV cache units are dequantized to a floating-point representation for further computation.
> > > > >
> > > > > > This dequantisation introduced computational overhead that slows down decoding, as mentioned in the related work on page 2.
> > > > >
> > > > > Yes. This dequantization is both time-consuming and memory-consuming. It takes time to convert the integer representation into a floating-point representation, and the dequantized KV cache units also require additional space on the GPU.
> > > > >
> > > > > > I also imagine dequantisation introduces additional memory as well, so would quantising the weights also not lead to potential memory and computation reduction?
> > > > >
> > > > > This is not always the case for memory. When conducting model inference with a quantized KV cache, dequantization is performed layer by layer; that is, when the decoding process reaches the $i$-th layer, only the KV cache of the $i$-th layer needs to be dequantized. Since the KV cache of a single layer in floating-point plus KV cache of other layers quantized is smaller than the full KV cache, there is a chance that the peak memory usage is reduced. However, in our Llama experiments, the full KV cache is already small due to the use of GQA, and the peak memory usage is mainly determined by the runtime hidden states. With KV quantization, the dequantized KV cache becomes the full KV cache (which is the dominating part of memory usage), so there is no notable memory reduction (or even an increase in memory usage due to system issues such as intermediate variables, GPU memory allocation, etc.). For compute, KV quantization does not lead to any reduction in computation.
> > > > >
> > > > >
> > > > > > Especially since it seems common to quantise and dequantise, while I would expect a fully quantised cache and model to remove the need for dequantisation and hence improve the memory footprint.
> > > > >
> > > > > Thanks for pointing this out. Indeed, quantizing everything can improve the memory footprint. Apart from KV cache quantization and weight quantization, activation quantization is vital in this circumstance. If the weight is quantized without hidden state quantized, dequantizing the KV cache is still needed to match the reprentation of hidden state. If everything is quantized (including the hidden states), this can be avoided.

---

### Author Response · Authors · 2025-11-04
**Manuscript Updated on November 4**

We thank the reviewers for their careful reading and constructive comments. We have completed the revision of the manuscript. The main updates are briefly summarized below.

Text marked in red corresponds to comments from Reviewer 4ga9, and text in blue corresponds to comments from Reviewer otMS.

Main revisions:

1. Added a background and preliminaries section (Section 2).

2. Added the Locret-Q variant in the method section (Section 4.4).

3. Reported variance in Table 2 and Table 3.

4. Fixed the inconsistency in the number of evaluated entries in Table 4 and Table 5.

5. Revised parts of the discussion to better match the experimental observations.

6. Added details on the generalizability of Locret (Appendix C).

7. Added an ablation study on the smoothing loss (Appendix D).

---

### Decision · Action_Editor_KXRL · 2025-12-09

**Recommendation:** Accept as is

**Audience:**

Yes

**Audience Explanation:**

Locret addresses an important  and relevant problem in efficient long-context LLM inference, a major bottleneck for LLM deployment. Long context inference becoming today an important aspect in particular with reasoning model. The method proposed is performant and lightweight to integrate. This make it relevant for TLMR audience.

**Claims And Evidence:**

Yes

**Claims Explanation:**

All reviewers agree that Locret is well-motivated, clearly described. While Reviewer 4ga9 identify some issue in particular some on the experimental side they were adressed during the rebuttal, making the current version of the paper empirically solid.